# Evidence accumulation, not 'self-control', explains dorsolateral prefrontal activation during normative choice

Cendri A Hutcherson[1,2]*[†], Anita Tusche[3,4][†]

[1]Department of Psychology, University of Toronto Scarborough, Toronto, Canada; [2]Department of Marketing, Rotman School of Management, University of Toronto, Toronto, Canada; [3]Division of Humanities and Social Sciences, California Institute of Technology, Pasadena, United States; [4]Departments of Psychology and Economics, Queen's University, Kingston, Canada

**Abstract** What role do regions like the dorsolateral prefrontal cortex (dlPFC) play in normative behavior (e.g., generosity, healthy eating)? Some models suggest that dlPFC activation during normative choice reflects controlled inhibition or modulation of default hedonistic preferences. Here, we develop an alternative account, showing that evidence accumulation models predict trial-by-trial variation in dlPFC response across three fMRI paradigms and two self-control contexts (altruistic sacrifice and healthy eating). Using these models to simulate a variety of self-control dilemmas generated a novel prediction: although dlPFC activity might *typically* increase for norm-consistent choices, deliberate self-regulation focused on normative goals should *decrease* or even *reverse* this pattern (i.e., greater dlPFC response for hedonistic, self-interested choices). We confirmed these predictions in both altruistic and dietary choice contexts. Our results suggest that dlPFC response during normative choice may depend more on value-based evidence accumulation than inhibition of our baser instincts.

*For correspondence:
c.hutcherson@utoronto.ca

†These authors contributed equally to this work

Competing interest: The authors declare that no competing interests exist.

## Editor's evaluation

This paper will be of interest to neuroscientists studying decision-making and the frontal lobe. On balance, the data provide more support for the view that the dorsolateral prefrontal cortex is involved in reading out the evidence in favor of different choice alternatives than the view that this region implements control processes that bias choices towards normative goals.

## Introduction

Self-control dilemmas typically involve tradeoffs between short-term, hedonic considerations and longer-term or more abstract standards and values. For example, social interactions often force an individual to weigh self-interest against norms favoring equity and other-regard. Similarly, dietary decisions often require weighing the immediate pleasure of consumption against personal standards or societal norms favoring healthy eating. Understanding when, why, and how people choose normatively preferred responses (e.g., generosity over selfishness, healthy over unhealthy eating, etc.) has represented a central goal of the decision sciences for decades. What neural and computational processes must be engaged to support more normative behavior? What makes such choices frequently feel so conflicted and effortful, and how can we make them easier? To what extent does following social or personal norms depend on activation in brain regions associated with inhibitory control, such as the dorsolateral prefrontal cortex (dlPFC)?

Previous research has provided a wealth of evidence suggesting that dlPFC activation is associated with normative choices in both the social and non-social domain. For instance, compared to unhealthy food choices, healthier choices in successful dieters were accompanied by greater activation in a posterior region of the dlPFC (*Hare et al., 2009*). Greater dlPFC response in a similar region has also been observed when individuals make normatively favored choices in both social decision making (*Strombach et al., 2015*; *Cutler and Campbell-Meiklejohn, 2019*) and intertemporal choice (*Luo et al., 2012*; *McClure et al., 2004*). Moreover, activation in the dlPFC increases when individuals explicitly focus on eating healthy (*Hare et al., 2011a*) or on decreasing craving for food (*Kober et al., 2010*). Electrical disruption of this area also decreases patience (*Figner et al., 2010*) and reduces normative behavior in cooperative contexts like the Ultimatum game (*Ruff et al., 2013*). Collectively, these results have been taken to support the notion that the dlPFC may be recruited to modulate values or bias choices in favor of normative responses, perhaps especially when those responses conflict with default preferences. Such findings are also consistent with the notion that this region might play a more general role in inhibitory control.

Yet, a variety of results seem inconsistent with this view. For example, researchers often fail to observe increased dlPFC recruitment when individuals make pro-social or intertemporally normative choices (*Zaki and Mitchell, 2011*; *Tusche et al., 2016*; *Magen et al., 2014*). Moreover, electrical disruption of the dlPFC has been observed both to *decrease* appetitive valuation of foods (*Camus et al., 2009*) and to *increase* generous behavior in altruistic choice tasks like the Dictator game (*Ruff et al., 2013*). Such findings conflict with the idea that this region consistently promotes normative concerns over immediate, hedonistic desires. Thus, it remains unclear how to predict whether and when one might observe a positive association between dlPFC response and choices typically associated with successful self-control.

Here, we propose a computational account of fMRI BOLD response in the dlPFC that may resolve many of these apparent inconsistencies. This account draws on prior research in both perceptual and value-based decision making, which consistently finds that the posterior dlPFC region associated with both normative 'self-control success' and inhibitory control tasks also activates during choices that are more difficult to discriminate in simple perceptual and value-based choices lacking a self-control conflict (e.g., *Heekeren et al., 2004*; *Noppeney et al., 2010*; *Pedersen et al., 2015*). Our account is also inspired by findings that the dlPFC may be one hub in a larger neural circuit – encompassing additional regions like the dorsal anterior cingulate cortex [dACC], supplementary motor area, and inferior frontal gyrus/anterior insula [IFG/aIns] – that selects actions for execution using a process of evidence accumulation toward a threshold for response (*Hanks et al., 2015*; *Hare et al., 2011b*; *Pisauro et al., 2017*; *Gluth et al., 2012*; *Rodriguez et al., 2015*). Based on this evidence, we developed a computational model of self-control dilemmas, rooted in the logic of evidence accumulation, that successfully predicts not only when an individual will choose in a normative rather than hedonistic (In the following results, we examine choices in two domains often associated with self-control:altruistic choice and healthy eating. We use the term 'normative' choices to describe generous and healthy decisions in these contexts, respectively. To contrast with normative choices, we refer to both selfish and unhealthy decisions using the term 'hedonistic' choices, although we acknowledge that decisions about money, even if selfish, would not be considered hedonistic in the classical sense of referring to sensory pleasure. Instead, we here use the term hedonistic in the sense of promoting immediate self-gratification. Similarly, we use the term 'hedonic attribute' to refer to attributes associated with immediate self-gratification (i.e., self-serving outcomes, food tastiness), whether or not they are directly linked to sensory pleasure.) fashion, but also when, why, and to what degree response in the dlPFC will be recruited during that process. We also note that we focus here on inhibitory control-related regions of the dlPFC rather than value-related dlPFC regions, which lie in somewhat more anterior areas of the lateral prefrontal cortex and frontal pole (*Tusche and Hutcherson, 2018*; *Plassmann et al., 2007*). Our model also applies in theory when observing similar relationships to other brain areas frequently associated with conflict and inhibitory control, including regions of the IFG/aIns and dACC.

Based on work examining simple perceptual and value-based choices (*Milosavljevic et al., 2010*; *Krajbich et al., 2010*; *Busemeyer et al., 2019*), we assume that the brain makes decisions through a process of value-based attribute integration and evidence accumulation (*Figure 1*). More specifically, choices are resolved by accumulating information about the value of choice-relevant information (attributes), weighted by their momentary goal relevance, until a critical threshold for choice is

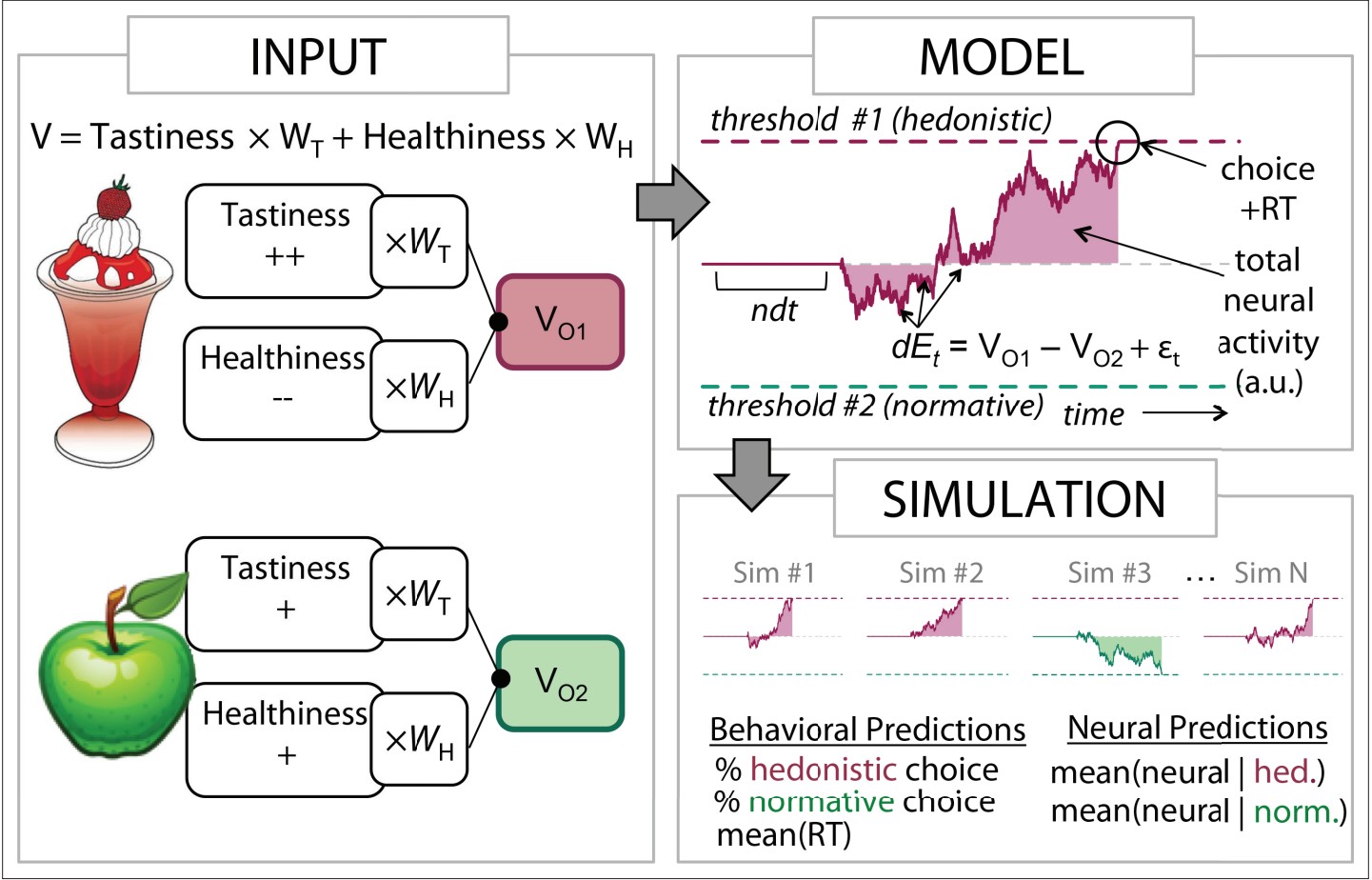

**Figure 1.** Attribute-based drift diffusion model (DDM) of normative choice. *Input.* Each option's hedonic and normative attributes (e.g., high tastiness and low healthiness for the sundae) are weighted by their current importance (e.g., $w_{Taste}$ [$w_T$] and $w_{Health}$ [$w_H$]) and summed to construct relative option values [$V_{O1} - V_{O2}$]. *Model.* In the computational model (upper right panel), these values, corrupted by momentary noise at time t [$\varepsilon_t$], serve as the inputs to an evidence accumulation process ($E_t$) representing the two options (Sunday vs. apple). This evidence accumulates over time, moving by an increment dE at each moment, until it hits one of two predefined thresholds, determining the simulated response time (RT), the simulated choice, and the simulated neural activity (the sum of the absolute value of the evidence from t=0 to t=RT). Choices are classified as normative if the option with higher normative attribute value is selected (in this example, higher healthiness, i.e., the apple in option 2). *Simulation.* Simulating this process thousands of times (lower right panel) allows distributions to be constructed for the expected frequency of hedonistic and normative choices, RTs, and average neural activity, mean(neural), for each type of choice.

reached. Some of these attributes are associated with hedonism (e.g., self-regarding concerns in altruistic choice), and some are associated with social norms and standards for behavior (e.g., other-regarding concerns). For expository purposes, we refer to these respectively as hedonic and normative attributes. Intuitively, whether a process of value-based evidence accumulation results in a hedonistic or normative choice depends not only on the magnitude of hedonic and normative attributes, but also on their weight: higher weights on normative attributes lead to more norm-consistent responses.

What role does the dlPFC play in the process of choice? The observation of increased posterior dlPFC response when people choose consistently with normatively favored goals (e.g., healthy over unhealthy choices) has been taken to suggest that this region acts either to modulate the processing of attribute values or their weights in favor of normatively favored goals (*Hare et al., 2009*; *Hare et al., 2011a*), or to inhibit hedonistic reward-related responding (*Lopez et al., 2014*; *Heatherton and Wagner, 2011*). In contrast, we propose that activity in this region reflects processes related to the *evidence accumulation stage* of decisions. This suggests that dlPFC response during normative choice represents a downstream consequence of valuation processes, rather than a direct causal influence upon them. To support this argument, we use a value-based drift diffusion model (DDM, *Figure 1*) to simulate when and why we might observe greater activity in the dlPFC (and regions with

similar response profiles) when resolving a choice. As we describe below, these simulations suggest that normative choices should be associated with greater neural activation in the dlPFC only when two things are true: hedonic attribute values *directly oppose* normative attribute values, and hedonic attributes receive *more weight* as inputs to the evidence accumulation process. In contrast, when normative attributes receive more weight, *hedonistic* choices should produce greater activity in the dlPFC and other areas associated with response selection.

We then used these observations to make two predictions. First, if people by default favor hedonic over normative attributes, then most studies will observe greater dlPFC response when people choose the normatively favored option. This prediction does not strongly distinguish our account from alternatives. However, our model makes a second, more novel prediction: if a normally hedonistic decision maker focuses on normative goals, this should *reduce* activation in the dlPFC when choosing the normatively favored option. A straightforward reading of an attribute-weighting account predicts the opposite: a normally hedonistic individual who deliberately attempts to focus on normative responding should show *increased* activation in the dlPFC in order to alter attribute weighting in favor of normative goals (*Lopez et al., 2014*; *Kelley et al., 2015*). We test these two alternative predictions across three datasets and two canonical self-control contexts in which people frequently struggle to align their actual behaviors with normative goals: altruistic and dietary choices. In all cases, results supported the predictions of an evidence accumulation account. These findings raise new and important questions regarding the role of the dlPFC – and effortful self-control more generally – in promoting normative choice.

## Results
### Simulating the dilemma of self-control

Although self-control dilemmas can take a variety of forms, for expository purposes, we here take a single, typical self-control dilemma: a decision maker deciding whether to indulge in a decadent snack (e.g., an ice cream sundae) or opt for something healthier (e.g., an apple). This example allows us to capture two critical features: first, self-control dilemmas typically involve making decisions about options that vary in the magnitude or value of hedonic and normative attributes (e.g., tastiness and healthiness). Second, the decision maker must weigh these attributes based on goals that can vary in their relative strength at different times and in different contexts. At a nice restaurant, tastiness may be prioritized. When trying to lose weight, healthiness is prioritized. We used simulations to explicitly capture these two features.

Simulations were realized using a value-based DDM (*Hare et al., 2011b*) in which an evidence signal capturing hedonic and normative attributes, weighted by their current subjective importance, accumulates toward a choice-determining threshold (see Methods for details). This model generated predictions for how *magnitudes* and *weights* for hedonic and normative attributes influence the likelihood of a virtuous (i.e., healthy) choice, response time (RT), and neural response. These simulations yielded three key observations about behavior and neural response, which we describe in the context of food choice but apply in theory across any self-control dilemma that requires weighing hedonic rewards against normative values and goals.

### Observation 1: The likelihood of a normative choice depends on the value of hedonic and normative attributes

To capture the idea that some choices (e.g., ice cream vs. Brussels sprouts) represent more of a self-control conflict than others (e.g., strawberries vs. lard), we simulated a single decision maker facing choices between hypothetical options that independently varied the relative value of normative and hedonic attributes (e.g., the foods' relative healthiness and tastiness). In the context of food choice, we classified a simulated choice as normative (healthy) when the simulation selected the option with higher normative attribute values (higher healthiness). Choices were classified as hedonistic (unhealthy) otherwise. To determine the effect of current behavioral goals, we simulated the decision maker's choices for a variety of different weights on healthiness ($w_{Health}$) and tastiness ($w_{Taste}$).

*Figure 2a* illustrates how variation in tastiness and healthiness of an option relative to the alternative affects a decision maker's *general* propensity to make a healthy choice (i.e., averaging over different instances of $w_{Taste}$ and $w_{Health}$). As can be seen, the magnitude and sign of the two attributes

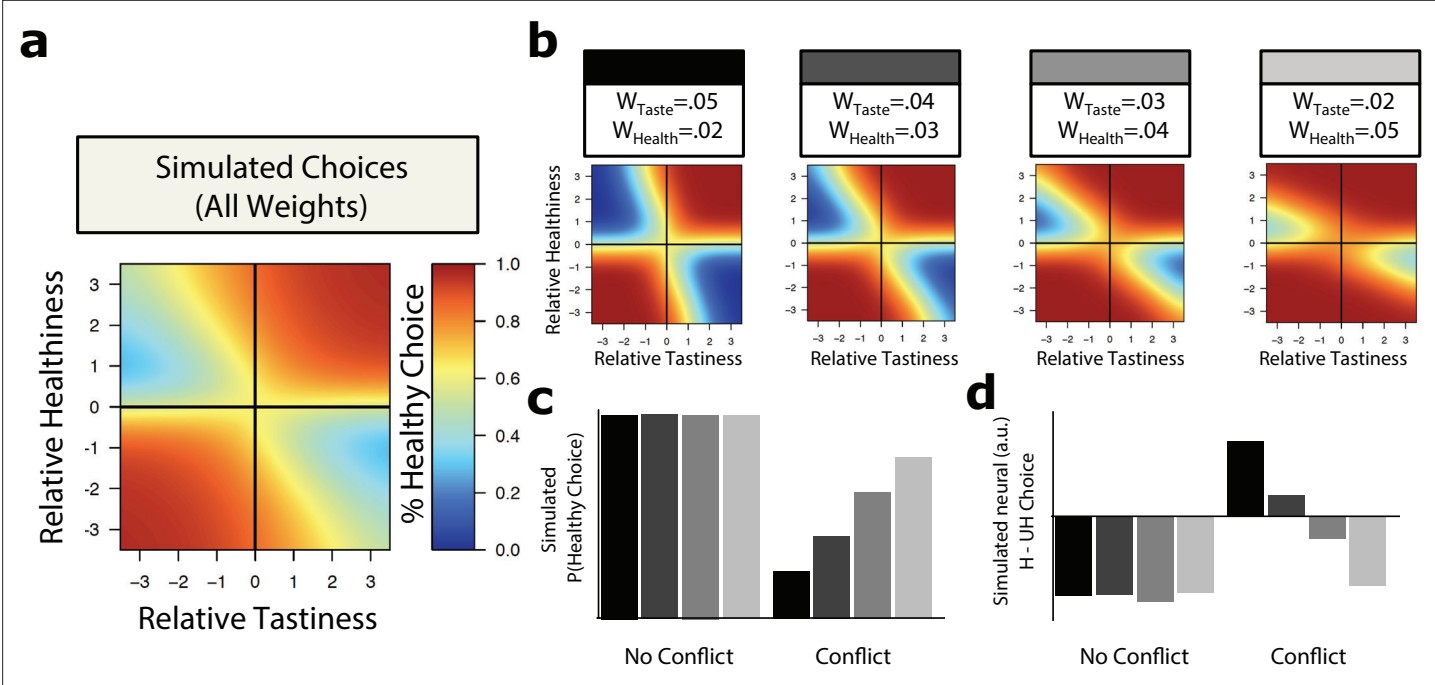

**Figure 2.** Simulating the dilemma of self-control. An evidence accumulation model (drift diffusion model [DDM]) can be used to simulate decision making for any self-control context requiring an integration of normative and hedonistic considerations (dietary choices displayed, integrating relative tastiness and healthiness). Heat maps display simulated healthy/unhealthy choices for binary choices consisting of a proposed option vs. a default option. Both options are characterized by attribute values such as the perceived tastiness and healthiness of the two food options, with plot axes capturing the relative value difference in tastiness (x-axis) and healthiness (y-axis) of the proposal vs. the default. For example, for each heat map, the upper right-hand quadrant shows likelihood of a healthy choice when the proposed option was both tastier and healthier than the default. The upper left-hand quadrant illustrates the likelihood of a healthy choice when proposed foods were healthier but less tasty than the default and so forth. Warmer colors indicate a higher likelihood of choosing the healthier option. (**a**) On average across multiple different goals (i.e., different weights on different attributes), the likelihood of a healthy choice depends on the relative attribute values of one option vs. another. Healthy choices are less likely when tastiness and healthiness conflict (conflict trials: one option is clearly better on one attribute, but worse on the other). (**b**) Specific goals prioritizing tastiness or prioritizing healthiness alter the frequency of healthy choice, although it still depends on the relative values of both tastiness and healthiness. Four example decision makers with different weights on healthiness and tastiness ($w_{Taste}$, $w_{Health}$) are displayed for comparison. (**c**) The overall likelihood of a healthy choice (averaged separately for all combinations of conflict or no-conflict choices) depends on attribute weights. Goals prioritizing tastiness (darker bars) produce fewer healthy choices than goals prioritizing healthiness (lighter bars), but only when tastiness and healthiness conflict. (**d**) The drift diffusion model can also simulate expected neural activity (i.e., aggregate activity summed over decision time: $\sum_{Time} |Evidence|$) when choosing healthy [H] or unhealthy [UH] options, as a function of relative attribute values of available options and different goals. Bars display the overall difference in neural activity for H compared to UH choices for goals prioritizing tastiness (darker bars) and healthiness (lighter bars), divided as a function of attribute conflict. In no-conflict trials, healthy choices elicit less activity regardless of goal (i.e., Activity $_H$ < Activity $_{UH}$). In conflict trials, however, healthy choices elicit more activity (i.e., Activity $_H$ > Activity $_{UH}$), but only when goals prioritize tastiness (dark gray bars). Identical results are obtained when substituting response time (RT) for neural response (see *Figure 2—figure supplement 1*). Similar results are also obtained when using a different formulation of the evidence accumulation process (see Appendix 2 and *Appendix 1—figure 1*).

The online version of this article includes the following figure supplement(s) for figure 2:

**Figure supplement 1.** Computational simulations of response time (RT).

matter: she tends to choose more healthily when one option dominates on both healthiness and tastiness (no-conflict trials). She chooses less healthily when one option is tastier while the other is healthier (conflict trials). She is least likely to choose normatively when the difference in tastiness is large, and the difference in healthiness is small. Thus, our simulations make the commonsense prediction that attribute values matter in determining the overall likelihood that an individual makes a healthy/normative choice.

## Observation 2: The likelihood of a normative choice depends on weights given to normative and hedonic attributes

We next attempted to capture the idea that individuals might vary in the goals that they prioritize, and also that the same individual might vary from context to context in those goals. In this case, the essence of self-control is to prioritize (i.e., assign a higher weight to) normative attributes like healthiness, or to deprioritize (i.e., assign a lower weight to) hedonic attributes like tastiness. We thus simulated the effects of different goal states by assuming different weights on hedonic and normative attributes (i.e., tastiness and healthiness). We show four example simulations in *Figure 2b*. Unsurprisingly, the decision maker chooses healthily less frequently when the weight on tastiness is higher than weight on healthiness. However, these differences are starkest in conflict trials, and essentially vanish for no-conflict trials (*Figure 2c*).

## Observation 3: Normative choices result in higher neural response only if attributes conflict and the decision maker weighs hedonic attributes more

The last and most important goal of our computational model simulations was to examine how neural response in a control region like the dlPFC (assuming its activity correlates with evidence accumulation) (*Figure 2d*). We characterized this simulated response as the absolute value of the accumulated evidence at each time point, summed over the duration of the choice, as this is what should contribute to observable BOLD responses.

Comparing differences in simulated neural response for healthy and unhealthy choices yields two important conclusions. First, when options do not conflict on healthiness or tastiness (i.e., one option is better on both), healthy choices generally elicit *less* activity than unhealthy ones (*Figure 2d*). Notably, for no-conflict trials, this holds true irrespective of whether a decision maker is currently prioritizing tastiness or healthiness. Second, and more importantly, when attributes *conflict*, neural activity during healthy vs. unhealthy choices shows a striking dependence on an individual's goals (i.e., the relative balance of $w_{Health}$ and $w_{Taste}$). In conflict trials, hedonism-favoring goals (i.e., $w_{Taste} > w_{Health}$) result in higher activity on average when choosing healthily. In contrast, when goals prioritize normative attributes like healthiness (i.e., $w_{Health} > w_{Taste}$), simulated neural responses are *lower* on average for healthy compared to unhealthy choices (*Figure 2d*). Thus, the neural response is positively associated with normative choice (i.e., greater neural activity to choose normatively instead of hedonistically) only when the decision maker places a higher weight on hedonistic than normative attributes. The same is true of simulated RTs, which are often used as a proxy for both choice difficulty and the presence of control (*Figure 2—figure supplement 1*). Thus, the logic of evidence accumulation models suggests that the observation that normative choices activate brain areas associated with cognitive control might simply indicate that hedonic attributes are currently weighed more highly.

## Testing computational predictions using fMRI data

### The DDM accurately predicts dlPFC activity across a variety of contexts

It is currently unknown whether activity in the dlPFC region frequently associated with inhibition and self-control might reflect activation patterns in the DDM during self-control. Preliminary evidence suggests that at least one region of the dlPFC correlates with evidence accumulation signals in simple choice (*Hare et al., 2011b*). We thus began by verifying that trial-by-trial simulated neural activity in the DDM correlated with activity in this region for complex, multi-attribute choices typical of different real-world self-control dilemmas. To assess the generalizability of this finding, we examined this across different choice contexts. We also show in supplementary analyses that this result does not depend on the particular form of the computational model but applies using an alternate formulations of a basic process of evidence accumulation to a threshold (see Appendix 2 for the example of an attribute-based neural DDM (anDDM) and *Appendix 1—figure 1*). Note, however, that finding a correlation within this region could occur because the dlPFC performs the precise computations carried out by the DDM, or could occur if the dlPFC performs separate computational functions that activate proportionally to evidence accumulation activity (e.g., RT). In either case, we would expect the trial-by-trial activity of the dlPFC to correlate with predictions of the DDM.

Our analysis focused on three previously collected fMRI datasets. Dataset 1[30] (N=51) and Dataset 2[22] (N=49) utilized an Altruistic Choice Task trading off different monetary outcomes for self and an

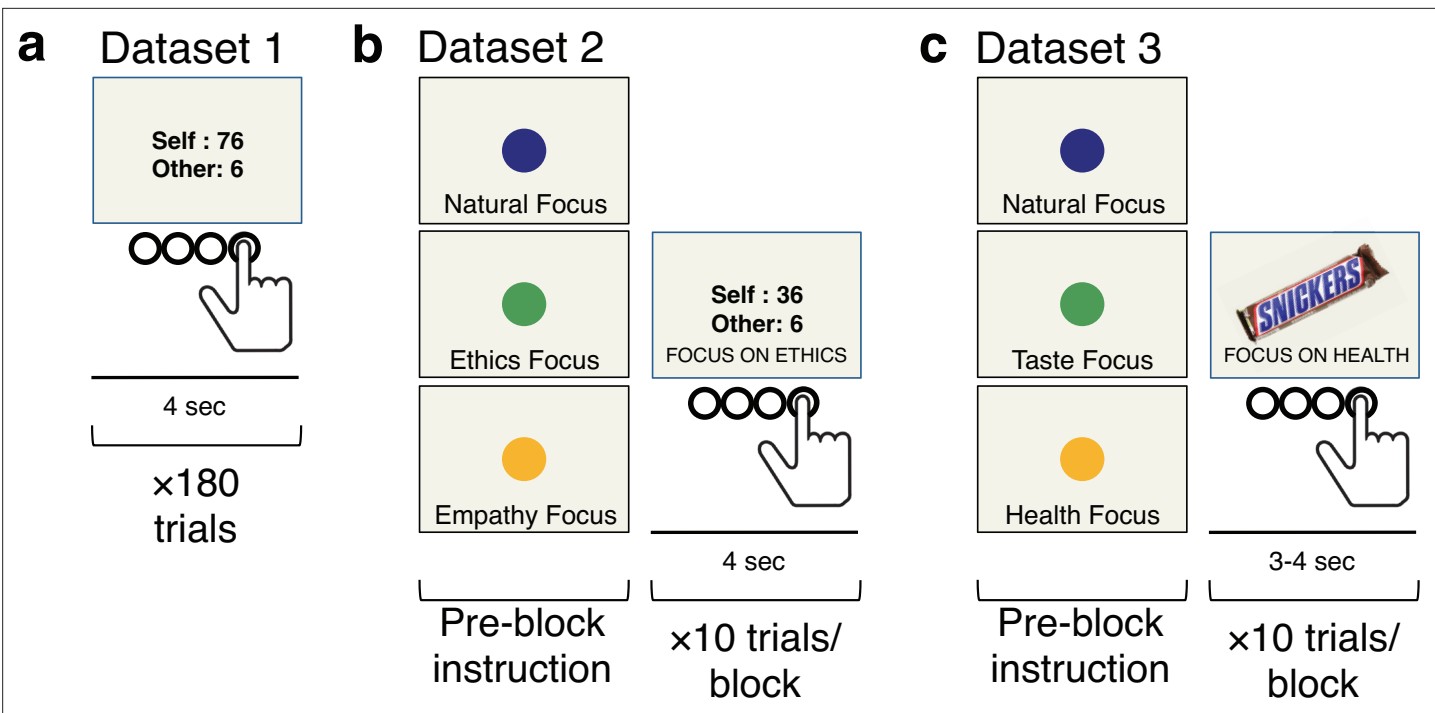

**Figure 3.** fMRI task designs. (**a**) In Dataset 1, participants made choices involving tradeoffs between monetary payoff for another person ($Other; normative attribute) and for themselves ($Self; hedonic attribute) in an Altruistic Choice Task. (**b**) In Dataset 2, participants made choices similar to the Altruistic Choice Task in Study 1, while we manipulated the *weights* on normative and hedonic attributes using instructions presented at the beginning of each task block. These instructions asked participants to focus on different pro-social motivations (ethical considerations, partner's feelings) as they made their choice. (**c**) In Dataset 3, we examined the generalizability of the model-based predictions in another choice domain. Here, participants performed a Food Choice Task in which the weights on food's healthiness (normative attribute) and tastiness (hedonic attribute) were manipulated. In all studies, participants had 4 s to decide, and gave their response on a 4-point scale from 'Strong No' to 'Strong Yes'.

anonymous partner in a modified version of a Dictator game (*Figure 3a and b*, see Methods for details). Dataset 3 utilized a Food Choice Task (*Figure 3c*), collected on three separate samples of participants (Sample 3a[22] N=36; Sample 3b[33] N=33[6]; Sample 3c[31] N=45, respectively), with different foods varying in tastiness and healthiness (see Methods for details). In Dataset 1 (altruistic choice), choices were made with the instruction to simply choose the most-preferred option (simulating natural choice contexts without experimentally manipulating goals/attribute weights). In Datasets 2 and 3, participants made choices in three separate conditions that manipulated goals/attribute weights by instructing participants to focus on different normative or hedonistic attributes (a point we return to below). All three datasets involved trials involving conflict between hedonic and normative attributes (conflict trials). Studies in Dataset 3 also included trials without such conflict, allowing us to test differential predictions of the computational model for conflict and no-conflict contexts (*Figure 2*). Because the three samples comprising Dataset 3 had an identical design, and neural results indicated no significant differences across the studies in regions of interest for the key analysis of normative vs. hedonistic choice, we report the results below collapsed across the samples (but see *Figure 4—figure supplement 2* for a comparison of results across samples).

We predicted that dlPFC activity should mirror the simulated activity of the DDM during self-control dilemmas. To avoid concerns about circular analysis and double dipping (*Kriegeskorte et al., 2009*), we tested the core predictions of the computational model using an independent dlPFC cluster as our primary region of interest (ROI). The selected dlPFC ROI was predictively associated with the term 'inhibitory control' in a text-based meta-analysis of functional neuroimaging studies (*Dockès et al., 2020*) (see Methods). However, to provide further evidence that neural activity in this dlPFC region is associated with predictions of the DDM, we also implemented supplementary analyses on the whole-brain level examining neural correlates of the DDM, both within and beyond the dlPFC. This analysis confirmed that activity in an area of dlPFC nearly identical to the area associated with inhibitory control consistently varies in the manner predicted by the DDM across several different choice

contexts (see *Figure 4a–c* for visualization of dlPFC regions correlated with DDM predictions, as well as Appendix 4 for further details of whole-brain results).

## Recruitment of the dlPFC when choosing normatively only occurs when goals are hedonistic and attributes conflict (Observation 3)

We first set out to confirm the central prediction of our simulations concerning the relationship between normative choices and activity in the dlPFC. In particular, models suggesting that the dlPFC promotes normative choices or inhibits prepotent responses (*Hare et al., 2009*; *Hare et al., 2011a*; *Heatherton and Wagner, 2011*) imply that norm-consistent choices should be accompanied by greater activation in the dlPFC (as has been observed previously). Moreover, this should be especially true when people focus on normative goals (*Hare et al., 2011a*; *Kober et al., 2010*), since those goals support norm-sensitive behavior and might require the override of default hedonistic preferences (*Lopez et al., 2014*; *Wagner et al., 2013*). The DDM makes the opposite prediction. While neural activity in the model (and by extension the dlPFC) *can* be higher for normative compared to hedonistic choices, this should be true only when goals lead to a stronger weighting of hedonic attributes and attribute values conflict (*Figure 2d*). Thus, if a regulatory focus on normative attributes increases their weight in the evidence accumulation process, this should increase normative choices, but result in *lower*, not higher, neural activity for those choices. We tested these predictions by performing an ROI analysis in the dlPFC, examining the contrast of activity for normative compared to hedonistic choices in different contexts. In Dataset 1 (altruistic choice), this involved choices made only during natural, unregulated decision making. In Dataset 2 (altruistic choice) and Dataset 3 (food choice), we examined choices made under different regulatory goals that were designed to increase or decrease weights on hedonic and normative attributes (i.e., payoffs for self and other in altruistic choice, tastiness and healthiness in food choice). As described above, we focused on an area predictively associated with the term 'inhibitory control' using the meta-analytic tool NeuroQuery (*Dockès et al., 2020*) (see *Figure 4a–c* [blue] for an illustration of the ROI). Results were similar when using other definitions of this ROI (see Appendix 3, 'Alternative ROI specifications' and *Appendix 2—figure 1*, *Appendix 3—figure 1*), as well as when focusing on other regions associated with the DDM at a whole-brain level (see Appendix 4 and *Appendix 4—figures 1 and 2*).

*Generous vs. selfish choices (Dataset 1)*. In Dataset 1, choices were defined as normative (i.e., generous) if the participant selected the option with less money for themselves and more money for their partner. Choices were defined as hedonistic (i.e., selfish) otherwise. Weights from the best-fitting model parameters indicated that subjects naturally placed more weight on their own outcomes (mean $w_{Self}$=0.0043 ± 0.0011 s.d.) than the other person's outcomes (mean $w_{Other}$ = 0.0018 ± 0.0018, paired-$t_{50}$=14.465, p<2.2 × 10$^{-16}$) or on fairness (i.e., |Self – Other|, mean $w_{Fairness}$ = 0.0011 ± 0.0032, paired-$t_{50}$=7.33, p=1.829 × 10$^{-9}$) (see *Supplementary file 1*). Given the higher weight on self-interest, a hedonic attribute, and the fact that all trials in this study involved conflict between normative and hedonic attributes, we predicted that we should observe greater neural response on average when people chose generously. We further predicted that individual differences in activation between generous and selfish choices should correlate *negatively* with individual differences in the relative weight on payoffs to other, a normative attribute, compared to payoffs for the self, a hedonic attribute. In other words, subjects who place more relative weight on the normative attribute should actually show *less* activation in the dlPFC during normative choice (see *Figure 4d* for both average simulated predictions and individual difference predictions based on the weights fitted to participants in Dataset 1). An ROI analysis of BOLD response for generous vs. selfish choices in the inhibitory control dlPFC ROI supported both predictions (*Figure 4g*). We observed a significant increase in average response during generous compared to selfish choices (paired-$t_{43}$=2.13, p=0.04, Cohen's d=0.32), confirming our prediction of higher dlPFC activity for normative choices. Moreover, for our primary dlPFC ROI, when correlating individual differences in weights on other vs. self (i.e., $w_{Other} - w_{Self}$) with differences in dlPFC BOLD response for generous vs. selfish choices, we observed a significant negative relationship of roughly the same magnitude as predicted by simulations (*r*=–0.31, p=0.04). A supplemental whole-brain analysis confirmed that this pattern was specific to the dlPFC, as well as regions of dACC and IFG/aIns that were also associated with the DDM, rather than a general property of neural activity (see *Appendix 4—figures 1 and 2*, and *Supplementary file 4* for details).

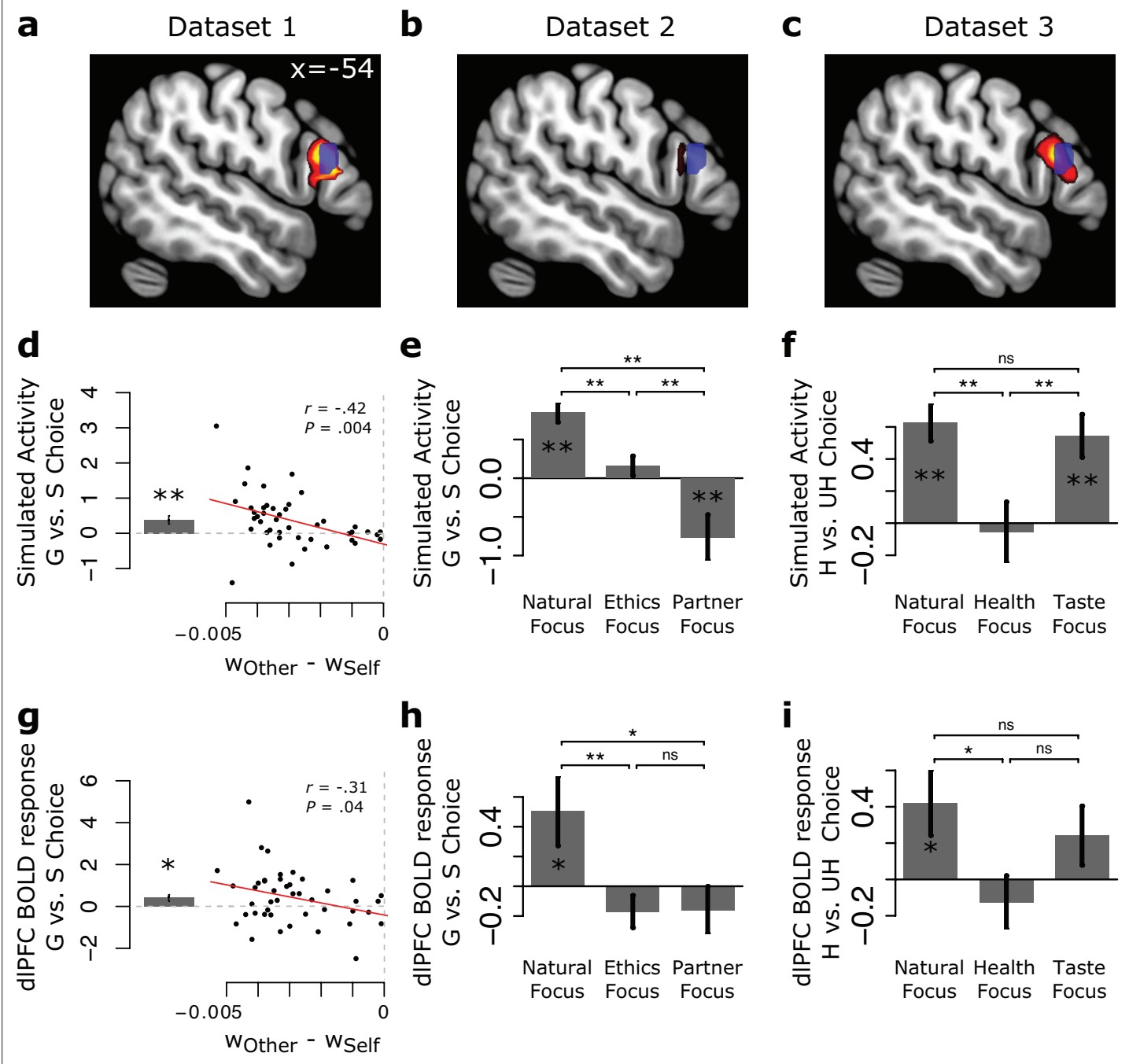

**Figure 4.** BOLD responses in the left dorsolateral prefrontal cortex (dlPFC) during self-control dilemmas. *Top (dlPFC):* Trial-by-trial BOLD response in the dlPFC correlates significantly with predicted activity of the drift diffusion model (DDM) across three separate datasets. This includes responses during altruistic choice (**a, b**) and dietary choice (**c**). Hot colors display DDM-related responses thresholded for display purposes at p<0.001 (**a and c**) and p<0.005 (**b**), uncorrected. The location of this activity was anatomically similar to an area of the dlPFC associated with inhibitory control in a meta-analysis (***Dockès et al., 2020***) (blue region of interest [ROI] displayed in each panel), which served as our primary, independently selected ROI to test predictions of the computational model (**d-f**). *Middle (Model Simulations):* Given participants' observed attribute weights, we simulated expected neural response during normative vs. hedonistic choice, using the DDM, for all trials in Dataset 1 (**d**), as well as separately for each condition in Dataset 2 (**e**) and Dataset 3 (**f**). *Bottom (dlPFC Results):* Within the independently defined dlPFC ROI associated with inhibitory control (blue voxels in **a–c**), BOLD response during normative choice vs. hedonistic choice when attributes conflict, in (**g**) Dataset 1 (N=50) for all trials, as well as in (**h**) Dataset 2 (N=49), and (**i**) Dataset 3 (N=113) as a function of regulatory goals. Regulatory goals were designed to decrease hedonic weights and increase normative weights on choice (Dataset 2: Ethics, Partner; Dataset 3: Health) or to emphasize weights on hedonic attributes (Dataset 3: Taste). Only conflict trials are considered across datasets. As predicted, normative choices *activate* the dlPFC, but only when goals result in a greater weight on hedonistic than

*Figure 4 continued on next page*

*Figure 4 continued*

normative attributes. Contexts that increase normative attribute weights yield *less* dlPFC activity during normative choices (Dataset 2: Ethics, Partner; Dataset 3: Health). *p<0.05 based on one-sample t-tests against 0, or paired-tests between conditions; **p<0.01. Error bars show standard error of the mean. Similar results were obtained from a left dlPFC region associated with dietary self-control in dieters (*Hare et al., 2009*), as well as a right-hemisphere version of the inhibitory control ROI (see Appendices 2-3 and *Appendix 2—figure 1* and *Appendix 3—figure 1* for details).

The online version of this article includes the following figure supplement(s) for figure 4:

**Figure supplement 1.** Model fits to behavior.

**Figure supplement 2.** Alternative visualization of model fits to behavior.

**Figure supplement 3.** Healthy vs. unhealthy choices by sample in Dataset 3.

*Regulatory effects on generous vs. selfish responding (Dataset 2)*. In Dataset 2 (also anonymous altruistic decision making and conflict trials only), we sought to replicate and extend these results. More specifically, we sought to test the DDM prediction that if regulatory goals increase the weight on normative attributes, this should result in *decreased* activation in the dlPFC when choosing normatively. Note that this prediction complements the negative correlation between dlPFC response and the balance of normative vs. hedonistic weights across subjects, but extends it to a within-subject prediction: even in participants who under normal circumstances might show greater activation in the dlPFC during generosity, interventions that shift weights in favor of normative attributes should reduce, rather than enhance, such differences. To manipulate weights on hedonic and normative attributes, we used an instructed cognitive regulation manipulation in which we asked participants on different trials either to 'Respond Naturally' (mirroring the natural preferences expressed by participants in Study 1) or to focus on one of two different goals ('Focus on Ethics' [Ethics], 'Focus on your Partner's Feelings' [Partner]) that both emphasize normative attributes, but in different ways (see Methods for details). To confirm that the manipulation influenced attribute weights, we performed one-way repeated-measures ANOVAs with condition (Natural, Ethics, Partner) as a fixed effect and best-fitting attribute weight parameters $w_{Self}$, $w_{Other}$, and $w_{Fairness}$ as dependent variables. This analysis confirmed that our manipulation yielded significantly different weights on the attributes across the conditions (all $F_{2,96} > 13.54$, all $p<6.59 \times 10^{-6}$, see Methods for details of model fitting). As expected, weights for self-interest (a hedonic attribute, $w_{Self}$) were highest in the Natural condition ($M_{Natural} = 0.0073 \pm 0.0035$ s.d.), lower in the Ethics condition ($M_{Ethics} = 0.0061 \pm 0.0047$), and lowest in the Partner condition ($M_{Partner} = 0.0037 \pm 0.0065$). By contrast, weights on the partner's outcomes and fairness (attributes related more strongly to social norms) increased with regulation ($w_{Other}$: $M_{Natural} = 0.0010 \pm 0.0038$, $M_{Ethics} = 0.0041 \pm 0.0045$, $M_{Partner} = 0.0051 \pm 0.0038$; $w_{Fairness}$: $M_{Natural} = 0.0017 \pm 0.0033$, $M_{Ethics} = 0.0053 \pm 0.0046$, $M_{Partner} = 0.0024 \pm 0.0035$) (see *Supplementary file 1* for more details on parameters).

Having confirmed that the regulatory focus manipulation altered weights on hedonic and normative attributes, we next asked if this manipulation affected BOLD response during generous vs. selfish choice in the dlPFC, consistent with predictions of the DDM. In particular, given that all trials involved conflict between normative and hedonic attributes, we predicted that in the Natural condition, where participants generally placed higher weight on self-interest (a hedonic attribute), *generous* choices should elicit higher activation. In contrast, in the Ethics and Partner conditions, which elicited more equal or even higher weight on normative attributes (i.e., other's outcomes and fairness), this difference should disappear or even reverse. In other words, the model predicts that in some cases, *selfish* choices should elicit the greatest activity in the dlPFC. Predictions of the DDM given the specific attribute weights observed in this study are illustrated in *Figure 4e*.

To test these predictions in the neural data, we performed one-way repeated-measures ANOVAs with condition (Natural, Ethics, Partner) as a fixed effect and average BOLD response in the dlPFC ROI for the contrast of generous vs. selfish choice as the dependent variable. This analysis revealed a significant effect of condition on dlPFC response ($F_{2,96} = 5.91$, p=0.004). Post hoc planned comparisons confirmed that in the Natural condition, generous choices elicited significantly greater activity in the dlPFC than hedonic choices ($t_{48}=2.16$, p=0.04, Cohen's d=0.31, *Figure 4h*), replicating the observed difference during Natural choices of Study 1. By contrast, in the Ethics and Partner focus conditions, generous choices no longer elicited significantly greater activation. Instead, *selfish* choices elicited *greater* activation, although the effect did not reach statistical significance. However, the key test of

our predictions was confirmed: the difference in activation between generous vs. selfish choice in both the Partner and Ethics conditions differed significantly from the difference in the Natural condition (G vs. S, Partner vs. Natural: $t_{48}$=2.57, p=0.01, Cohen's d=0.36; G vs. S, Ethics vs. Natural: $t_{48}$=3.13, p=0.003, Cohen's d=0.45). Thus, in the same individuals, the association between generous choices and *higher* activation in the dlPFC depended on whether goals emphasized selfishness rather than social norms (*Figure 4h*). Supplemental whole-brain analyses and examination of individual differences confirmed these findings (see *Supplementary file 4* and Appendix 5 for details).

*Regulatory effects on healthy vs. unhealthy choice (Dataset 3)*. In Dataset 3, we sought to replicate the finding that a regulatory focus on normative attributes reduces activation in the dlPFC in a different, non-social domain: healthy eating. During the Food Choice Task in Dataset 3, attribute weights were manipulated by instructing participants either to 'Respond Naturally', 'Focus on Health', or 'Focus on Taste' while making their choice. Normative (i.e., healthy) choices were defined as selecting the food with higher subjectively perceived healthiness (see Methods for details). Note that the 'Focus on Health' instruction aimed to increase the weight on healthiness ($w_{Health}$), a normative attribute. Extending results in Dataset 2, the 'Focus on Taste' condition was designed to enhance the weight on tastiness ($w_{Taste}$), the hedonic attribute, which should preserve or even enhance the difficulty of normative choices that we observed in natural choice settings in Datasets 1 and 2. This allowed us to verify that our findings are specifically driven by changes in weights, not simply because we asked participants to perform a cognitive task.

To confirm that the regulatory manipulation influenced attribute weights, we performed one-way repeated-measures ANOVAs, similar to Dataset 2, with condition (Natural, Taste, Health) as a fixed effect and estimated attribute weight parameters $w_{Taste}$ and $w_{Health}$ as dependent variables. Because Dataset 2 combined data from three individual participant samples (see Methods for details), we also included Study as an additional factor in the analysis. Here, we report the effect of the manipulation on attribute weights across all studies (*Supplementary file 1*, but see *Supplementary file 2* for results in each sample separately). This analysis confirmed that our manipulation yielded significantly different weights on the different attributes across the conditions (all $_{F2,220}$ > 44.43, all p<2.2 × 10$^{-16}$). As expected, weights on tastiness (a hedonic attribute) were highest in the Taste and Natural conditions ($M_{Taste}$ = 0.0056 ± 0.0043, $M_{Natural}$ = 0.0058 ± 0.0036), and lowest in the Health condition ($M_{Health}$ = 0.0027 ± 0.0033). Weights on healthiness (a normative attribute) showed the opposite pattern, being lowest in the Taste condition ($M_{Taste}$ = 0.0002 ± 0.0022), similar though slightly higher in the Natural condition ($M_{Natural}$ = 0.0005 ± 0.0021), and highest in the Health condition ($M_{Health}$ = 0.0042 ± 0.0035).

Next, we generated predictions for the pattern of dlPFC response on the subset of trials involving conflict. For each condition, we used each individual's best-fitting weights to simulate neural responses using the DDM (*Figure 4f*). These simulations predicted that healthy compared to unhealthy choices should on average elicit the greatest activation in the dlPFC in the Taste and Natural conditions (though unexpectedly, the model predicted a non-significantly *smaller* difference in Taste trials). More importantly, the model predicted that this difference should be significantly smaller in the Health condition. Because observed weights on tastiness and healthiness in the Health condition were similar in strength, the within-condition comparison between healthy and unhealthy choices was not predicted to be significant. To test these predictions, we performed a one-way repeated-measures ANOVA, similar to Dataset 2, with condition (Natural, Taste, Health) as a fixed effect and the average dlPFC BOLD response in the contrast of healthy vs. unhealthy choice (limited to trials with attribute conflict) as the dependent variable. We also included sample (i.e., Sample 3a, 3b, 3c) as a factor interacting with condition, but observed no differences in dlPFC activity as either a main effect of sample ($F_{2,101}$ = 2.15, p=0.12) or as a modifier of condition effects ($F_{4,177}$ = 1.52, p=0.19). We thus report results collapsed across studies (but see *Figure 4—figure supplement 2* for visualizations of results separated by study). As predicted by the simulations (*Figure 4f*), this analysis revealed a significant effect of condition on dlPFC response ($F_{2,177}$ = 4.11, p=0.02). Follow-up t-tests confirmed the predicted direction of dlPFC activation (*Figure 4i*). BOLD response during healthy compared to unhealthy choices was significantly greater in the Natural condition for the dlPFC (paired-$t_{100}$=2.35, p=0.02, Cohen's d=0.24, within-condition comparison). As predicted, in the Health condition, healthy vs. unhealthy choices elicited comparable dlPFC responses (paired-$t_{97}$=0.86, p=0.39, Cohen's d=0.09, within-condition comparison) and even showed a trend toward greater activity for *unhealthy* choices. Importantly, the reduction of the difference in activity during healthy vs. unhealthy choices between

Natural and Health conditions was significant (paired-$t_{93}$=2.58, p=0.01, Cohen's d=0.27). Response for healthy vs. unhealthy choice in the Taste condition lay in between these two extremes, as predicted by the model. While the comparison of healthy vs. unhealthy choices in the Health vs. Taste condition was not significant (paired-$t_{97}$=1.35, p=0.18, Cohen's d=0.15), we attribute this discrepancy to the fact that the Taste condition had very few healthy choices per individual. Thus, the comparison of healthy vs. unhealthy choices was significantly noisier to estimate, and based on a smaller number of subjects (N=84). Nevertheless, our primary hypothesis was confirmed: in the same individuals, healthy choices could be accompanied by *higher* activation in brain regions typically associated with cognitive control (when goals naturally emphasized hedonism), or *matched* or even *lower* activation (when goals emphasized health norms). Supplemental whole-brain analyses confirmed that this pattern of results was specific to the dlPFC and other regions associated with the DDM (see *Appendix 4—figures 1 and 2*, and *Supplementary file 4* for details).

## Absence of regulatory effects in the absence of conflict (Dataset 3)

Our analyses so far focused on conflict trials, since simulations suggest that these trials show the biggest differences as a function of attribute weights (*Figure 2*). The design of Dataset 3, which included a subset of trials with no attribute conflict, also allowed us to test one further prediction of the DDM. In Observation 3, we found that normative choices should be associated with increased neural activity *when hedonic and normative attributes conflict* but the reverse when they do not (*Figure 2d*). Moreover, the DDM suggests little to no differences in response across goal contexts favoring hedonism or health norms. This suggests that, in contrast to conflicted choices, there should be less effect of regulatory focus on dlPFC response during no-conflict choices.

To test this prediction, we first performed a one-way repeated-measures ANOVA with condition (Natural, Taste, Health) as a fixed effect and the average BOLD in the dlPFC for the contrast of healthy vs. unhealthy choice as the dependent variable, focusing only on the subset of trials with no-conflict between tastiness and healthiness of a food (i.e., when the value of the option was positive or negative for both). As before, we also included sample (3a, 3b, 3c) as a factor in this analysis. Because we found a non-significant difference across studies in dlPFC response during unconflicted trials ($F_{4,209}$ = 2.26, p=0.06), we report the results for Dataset 3 below collapsed across samples. As predicted, when examining only non-conflict trials, there was no significant influence of regulatory condition on the difference in neural activity between healthy and unhealthy choice ($F_{2,213}$ = 0.357, p=0.7). Given this lack of effect across conditions, we averaged the three conditions together to analyze the main effect of healthy vs. unhealthy choice. This analysis indicated that healthy choices were accompanied by a non-significantly *lower* response in this region (paired-$t_{104}$=0.21, p=0.84, Cohen's d=0.02). In other words, as expected from model simulations (*Figure 2d*), activation in the dlPFC for normative choices when normative and hedonic attributes did not conflict is generally low, and shows no significant effect of regulatory focus.

## Discussion

When and why do normative choices – which conform to abstract standards and social rules – recruit regions associated with cognitive control like the dlPFC? Simulated activity from an attribute-based DDM suggests a straightforward answer: normative behavior may only trigger the dlPFC when normative attributes conflict with hedonic ones, and the decision maker values hedonic attributes more. Across three separate fMRI datasets and two different choice domains (generosity and healthy eating), we show several results that confirm predictions of the DDM. First, we show that activation in the dlPFC correlates consistently with predicted neural responses of the DDM across all contexts examined. Second, we show that even in individuals who show a natural bias toward selfishness, regulatory instructions to focus on socially normative attributes increase generosity, but *reduce* dlPFC response when choosing generously. Third, this pattern replicated in the domain of healthy eating, suggesting a general principle that may apply across a variety of self-control dilemmas. Our results provide empirical support for recent theories positing that successful self-control – defined as choosing long-term or abstract benefits over hedonic, immediate gratification (*Duckworth, 2011*) – depends importantly on value computations. They stand in contrast to the predictions of models of posterior dlPFC function suggesting that the strength with which the dlPFC activates during choice determines whether

prepotent hedonistic responses are resisted (*Lopez et al., 2014*; *Kelley et al., 2015*; *Wagner et al., 2013*; *Hofmann et al., 2009*). Our results point to a modified conceptualization of the role played by the dlPFC in promoting normative choice.

A large literature, generally consistent with models that assume normative behavior requires controlled processing, suggests that the dlPFC activates when prepotent responses conflict with desired normative outcomes (*Aron et al., 2004*; *Garavan et al., 1999*). The neural activity in the DDM, which arises from evidence accumulation signals, is in some ways consistent with such an interpretation. However, it calls into question assumptions that prepotency equates to hedonism, or even to automaticity (*Bargh, 1994*) more generally. Instead, our model suggests that the 'prepotent response' may correspond, at least in the realm of value-based decision making, to choices consistent with the choice attribute that is currently receiving higher weight, *regardless of the source of that weight*. In other words, even when higher weights on normative attributes derive primarily from a deliberative, regulatory focus, as in our final two datasets (*Hare et al., 2011a*; *Tusche and Hutcherson, 2018*; *Schmidt et al., 2018*), this yields *reduced* activity in the dlPFC when making normative choices. Mechanistically, these patterns result from the fact that higher weights on normative attributes reduce the computation required for neural accumulation signals to trigger a normative response. Thus, while virtuous choices associated with successful self-control may sometimes recruit the posterior dlPFC, manipulations that increase the weight on normative attributes, either by making it more salient in the exogenous environment or focusing endogenous attention toward it, should both promote normative behavior and make it *easier* to accomplish.

This observation may help to explain why some researchers have found evidence consistent with greater response in the dlPFC promoting normative choice (*Hare et al., 2009*; *Cutler and Campbell-Meiklejohn, 2019*; *Hare et al., 2011a*; *Ruff et al., 2013*; *Hakimi and Hare, 2015*; *Baumgartner et al., 2011*), while others have not (*Zaki and Mitchell, 2011*; *Tusche et al., 2016*; *Magen et al., 2014*). Variations that influence the weight on normative attributes – whether across individuals, goal contexts, or paradigms – will tend to reduce statistical significance and increase heterogeneity in the link between neural activation in the dlPFC and normative choice. Fortunately, our model provides a way to predict both *when* and *why* dlPFC activity will be observed. For example, in the domain of intertemporal choice, our model predicts that making future outcomes more salient should amplify their weight in the choice process, promoting patience while *decreasing* dlPFC activation. This is exactly what is observed empirically (*Magen et al., 2014*). Thus, researchers would do well to interpret activation of the dlPFC for a particular kind of choice (be it generous or selfish, healthy or unhealthy, patient or impatient) with caution. Such a pattern may say less about whether the dlPFC (and by extension, cognitive control more generally) is *required* to inhibit instinctual responses and preferences, and more about what kinds of attributes are most salient or valuable at the moment.

Our results have important implications for theories of self-control suggesting that the dlPFC promotes self-control by modulating attribute weights in the choice process (*Hare et al., 2009*; *Baumgartner et al., 2011*; *Rudorf and Hare, 2014*). The region of dlPFC that correlated with the DDM in our datasets is nearly identical to areas previously found when dieters made healthy compared to unhealthy choices (*Hare et al., 2009*), and when participants were required to recompute values based on contextual information (*Rudorf and Hare, 2014*). Yet, we find that the relationship between self-control 'success' and 'failure' in this region attenuates when participants actively focus on health: dlPFC now responds non-significantly more strongly to *unhealthy* choices. These results thus seem incongruent with the notion that this area down-regulates weight on norm-inconsistent considerations and up-regulates norm-consistent ones, since we observed *decreased* responses in this area in the context of *increased* normative choice and *increased* weight on normative attributes (*Figure 4*, c.f. *Tusche and Hutcherson, 2018*). Instead, this region appeared to correlate with the evidence accumulation stage of decisions, rather than with the evidence construction stage, responding during decision conflict generally, regardless of whether that conflict derived from greater weighting of hedonic or normative attributes.

We emphasize, however, that our results and conclusions apply narrowly to the area of dlPFC examined here. The evidence accumulation-related dlPFC region in this study, which also corresponds to the area identified by the meta-analytic 'inhibitory control', lies posterior and dorsal to another dlPFC area that we have observed, in these same datasets, to track hedonistic and normative attributes in a goal-consistent manner and to serve as a candidate for mediating regulation-induced changes

(*Tusche and Hutcherson, 2018*). Furthermore, gray matter volume in this more anterior dlPFC area, but not in the posterior dlPFC region identified here, correlates with regulatory success (*Schmidt et al., 2018*). Thus, while some areas of the dlPFC may indeed play an important role in promoting self-regulation and normative behavior by altering attribute weights in decision value, we suspect that they are anatomically and computationally distinct from the region of the posterior dlPFC sometimes assumed to serve this role. Future work will be needed to better delineate subregions of the dlPFC, and to determine the unique role each one plays in promoting normative choices.

The correspondence between predictions of the DDM and activation patterns in the dlPFC makes it tempting to conclude that this region performs this computational function. While this hypothesis is consistent with results from single-cell recordings (*Hanks et al., 2015*; *Hunt et al., 2018*), we also acknowledge that the dlPFC has been associated with many computational functions and roles, not all of which are mutually incompatible. Thus, it is possible that the dlPFC region observed here performs some sort of process that is correlated with, but not identical to, the neuronal computations of the DDM. Indeed, although we found that dlPFC responses mostly conformed to predictions of the DDM, we also found a few notable divergences. For example, during unconflicted trials, the DDM predicts that *hedonistic* choices should be strongly associated with greater activity than normative choices. While we observed a weak effect in this direction in Dataset 3 (the only dataset in which this prediction could be tested), it was not significant. Similarly, in Dataset 2, the DDM predicted that the comparison of generous vs. selfish choices should be lower when focused on one's partner compared to when focusing on ethics, while the empirical data showed no difference. While some of these discrepancies could be due simply to the noisiness inherent to fMRI (especially in studies that were originally designed to address other questions), it is also possible that they serve as clues that the computational function of the dlPFC may be strongly correlated with, but importantly distinct from, the evidence accumulation process itself. Future work, including computational modifications or additions to the DDM, and recordings from other modalities like EEG or intracranial recordings (*Hunt et al., 2018*), may help to elucidate the precise computational functions served by this area and the ways in which it promotes adaptive choice and normative behavior. Our work serves to highlight the value of computational models for understanding the neural and psychological computations involved in self-control (*Tusche and Bas, 2021*). Work extending these findings to other domains of normative choice, such as moral decision making or intertemporal choice, may also help to identify the commonalities and differences across different self-control dilemmas.

## Methods
### Computational model simulations

Our attribute value-based DDM (*Figure 1*) assumes that brain areas involved in decision making (particularly those that convert preferences into action) demonstrate activity that correlates with an evidence accumulation process shaped by the weighted value of attributes on each trial. A choice results at time t', the first moment at which the accumulated evidence signal EA exceeds one of two predetermined thresholds or barriers ±B. The total RT is t' plus a constant non-decision time (ndt) that accounts for perceptual and motor delays. The instantaneous evidence accumulation signal $EA_t$ represents the amount of information favoring choice of one option over another at time t and is captured by the equation

$$EA_t = EA_{t-1} + \sum_i \left( w_i Attrib_i^1 - w_i Attrib_i^2 \right) + C + \varepsilon_t$$

where $Attrib_i^1$ represents the value of option 1 for attribute i, C represents a constant bias in the drift which is positive (negative) for consistent biases toward option 1 (option 2), and $\varepsilon_t$ represents instantaneous Gaussian noise distributed as $\sim N(0, 0.1)$ in simulations. In our simulations, we assumed two independent attributes: one related to hedonism (e.g., tastiness of a food) and one related to norms and standards (e.g., healthiness), although in principle any number and type of attribute could occur. Using these equations allowed us to simulate the dynamically evolving evidence accumulation signal, and to derive distributions of both RTs and neural response, which we assumed was proportional to the total accumulated evidence summed over the time of the decision (i.e., $\sum_t |EA_t|$). We label the

final output (i.e., choice) of the system as 'normative' if it results in selecting the option with the higher unweighted value for the normative attribute (e.g., the option with higher healthiness).

To simulate everyday self-control dilemmas using this framework, we simulated choices between two options representing different combinations of hedonistic and normative attributes, allowing the relative value difference between an option and its alternative on a given attribute to vary independently in the arbitrarily chosen range [-3,–2, …+2,+3]. This permitted us to explore how the likelihood of a normative choice changes depending on how much better or worse one of the two options is along hedonic and normative attribute dimensions, as well as what happens when the relative values of the two attributes conflict (i.e., take opposite signs) or do not.

Our simulations also sought to capture the notion that a decision maker can vary from moment to moment in their commitment to and desire for hedonistic vs. normative goals. For example, a dieter may begin to relax the importance they place on norm-consistent attributes like healthiness once they reach their target weight, resulting in more unhealthy choices. In the main text (and *Figure 2*), we describe simulations for four different goal contexts with different asymmetric combinations of weights: two had higher weights on tastiness, a hedonic attribute (i.e., $w_T = 0.05$, $w_H = 0.02$, and $w_T = 0.04$, $w_H = 0.03$) and two with higher weights on healthiness, a normative attribute ($w_T = 0.03$, $w_H = 0.04$, and $w_T = 0.02$, $w_H = 0.05$). For simplicity, we assumed that all choices used a choice-determining threshold $\pm B = 0.15$, selected to produce RTs in the range typically observed in human subjects. Thus, for illustration purposes, we simulated a decision maker in four different contexts with different weights on the two attributes, facing 49 distinct choices representing different combinations of attribute values. To ensure that our conclusions held across a variety of weights, we also simulated an additional 32 different goal contexts, fully covering the factorial combination of weights on $w_T$ and $w_H$ in the range of 0–0.05 (0.01 increments). Using these values and weights, we simulated hedonistic and normative choice frequencies, total neural activation, and RTs for each of the different hypothetical option pairs/attribute combinations, probing the effects of attribute weights, attribute magnitudes, and attribute conflict (i.e., match or mismatch between the signs of normative and hedonic attribute). The results of these simulations are displayed in *Figure 2*. Code is available at https://osf.io/eqvwd/.

## Experimental datasets

To test predictions of the DDM, we re-analyzed existing choice task data. Details about portions of Datasets 1, 2, and 3, as well as neuroimaging parameters, have been reported previously (*Hare et al., 2011a*; *Tusche and Hutcherson, 2018*; *Hutcherson et al., 2015*; *Schmidt et al., 2018*). Here, we highlight in brief the most important details for the current work.

### Participants

For Dataset 1[30], we analyzed data from a study with 51 male volunteers (mean age 22). For Dataset 2[22], we analyzed data from a study with 49 volunteers (26 male, mean age 28, range 19–40). For Dataset 3, we combined data from three studies using a dietary choice task (Study 3a[22] N=36 individuals, mean age 29 from Dataset 2 returning to the lab for a separate session on a separate day; Study 3b[6] N=33 individuals, mean age 24.8; Study 3c[31] N=44 individuals, mean age 24.8). Caltech's Internal Review Board approved all procedures for Dataset 1, Dataset 2, and Studies 3a and 3b of Datasets 3. The Internal Review Board at INSEAD approved all procedures for Study 3c. Participants in all studies provided informed consent prior to participation.

### Tasks and stimuli

*Altruistic Choice Task (Datasets 1 and 2)*. We examined self-control dilemmas pitting self-interest against generosity using an Altruistic Choice Task for Datasets 1 and 2. On every trial in the fMRI scanner, the participant chose between a proposed pair of monetary prizes for herself and a real but anonymous partner, or a constant default prize-pair for both ($50 in Dataset 1, $20 in Dataset 2) (*Figure 3a–b*). Proposed prizes in the prize-pair varied from $0 to $100 in Dataset 1 and $0 to $40 in Dataset 2, and always involved one individual receiving an amount less than or equal to the default, while the other individual received more. Thus, on every trial, the participant had to choose between generous behavior (benefitting the other at a cost to oneself) and selfish behavior (benefitting oneself at a cost to the other).

Upon presentation of the proposal, participants had up to four seconds to indicate their choice using a 4-point scale (Strong No, No, Yes, Strong Yes), allowing us to simultaneously measure both their decision and strength of preference at the time of choice. The direction of increasing preference (right-to-left or left-to-right) varied for each round of the task in Dataset 1, and across participants in Dataset 2. If the subject did not respond within 4 s, both individuals received $0 for that trial.

To increase the anonymity of choices, the participant's choice was implemented probabilistically: in 60% of trials, she received his chosen option, while in 40% of trials, her choice was reversed and she received the alternative, non-chosen option. This reversal meant that while it was always in the participant's best interest to choose according to her true preferences, her partner could never be sure about the actual choice made. Probabilistic implementation does not strongly influence the choices participants make (*Tusche and Hutcherson, 2018*; *Hutcherson et al., 2015*), but permits more plausible anonymity, increasing the self-control challenge involved in choosing generously. The participants were informed that the passive partners were aware of the probabilistic implementation, and the outcome was revealed on every trial 2–4 s following the response. At the end of the task, one trial was randomly selected and implemented according to the participant's choice in the selected trial.

Dataset 1 included 180 trials total, with no specific instructions for how to respond. Dataset 2 included 270 trials, 90 each in three instructed focus conditions (*Figure 3b*). See the *Manipulating normative goals (Datasets 2 and 3)* section below for details on these instructions.

*Dietary Choice Task (Dataset 3)*: We examined self-control dilemmas in a second context pitting hedonism against healthy eating using a Dietary Choice Task for Dataset 3. Sample 3a was collected in our laboratory and completed on a subset of participants from Dataset 2. Samples 3b and 3c were collected in other research labs, but had nearly identical structure. We thus describe common elements among all tasks, highlighting minor differences where relevant.

Prior to the main choice task, participants rated a set of different foods for their healthiness and tastiness (200 foods in Study 3a, 180 in Studies 3b and 3c). For Study 3a, these ratings were used to select a pool of 90 foods that covered a range of health and taste ratings. For all three studies, they were used to select a neutral reference food rated as neutral on both health and taste. This neutral food served as individuals' default option for each choice (similar to the monetary default option in the Altruistic Choice Task).

On each trial in the fMRI scanner (270 trials in Study 3a, 180 in Studies 3b–c), participants saw one selected food (*Figure 3c*) and had to decide whether they would prefer to eat the displayed food or the reference food. As in the Altruistic Choice Task, participants had up to 4 s in Study 3a and 3 s in Studies 3b–c to indicate their choice using a 4-point scale (Strong No, No, Yes, Strong Yes). If the subject did not respond within the time limit, one of the foods was selected randomly. To match the instructed goal manipulation used in the Altruistic Choice Task, participants completed trials each in one of three instructed focus conditions (90 trials per condition in Study 3a, 60 trials per condition in Studies 3b–c). See the *Manipulating normative goals (Datasets 2 and 3)* section below for details. In all three studies, one trial was randomly selected at the end and implemented.

In Study 3a, to match the probabilistic outcome used in the Altruistic Choice Task, the participant's choice was also implemented probabilistically in the Food Choice Task. In 60% of trials, participants received the chosen option, while in 40% of trials the choice was reversed and she received the alternative, non-chosen option. To reduce the length of the task, participants did not see this outcome on every trial. Instead, three trials were selected randomly at the end of each scan, and participants viewed their choice as well as the probabilistic outcome on that trial. In Studies 3b–c participants simply received their choice on a randomly chosen trial at the end of the experiment.

## Manipulating normative goals (Datasets 2 and 3)

Our computational model simulations suggested that the extent to which normative choices are associated with greater neural response depends to a large extent on the priority or weight given to normative vs. hedonic attributes. We thus capitalized on the design of Datasets 2 and 3, which manipulated attention to different attributes (and corresponding weights), allowing us to test specific predictions of the DDM.

## Manipulation of altruistic goals (Dataset 2)

To manipulate attention to different attributes, during the Altruistic Choice Task in Dataset 2, participants completed trials in three instructed focus conditions: Respond Naturally, Focus on Ethics, and Focus on Partner. During *Natural* trials, participants were told to allow whatever feelings and thoughts came most naturally to mind, and to just choose according to their preferences on that trial. During *Ethics* trials, participants were asked to focus on doing the right thing during their choices. They were encouraged to think about the justice of their choice, as well as its ethical or moral implications, and to try to bring their actions in line with these considerations. During *Partner* trials, participants were asked to focus on their partner's feelings during their choices. They were encouraged to think about how the other person would be affected, as well as whether they would be happy with the choice, and to bring their actions in line with these considerations.

Each participant completed 90 trials per condition, presented in randomly interleaved blocks of 10 trials. A detailed set of instructions informing participants of their task for the upcoming block of trials was presented for 4 s prior to the block, and participants were asked to focus on the specific instruction for all trials within that block.

## Manipulation of health goals (Dataset 3)

Analogous to the Altruistic Choice Task in Dataset 2, healthy eating was manipulated in three experiments (Studies 3a–c) using an instructed focus manipulation. Each participant completed all three attentional conditions (90 trials per condition in Study 3a, 60 trials per condition in Studies 3b–c, randomly interleaved blocks): *Natural* Focus, *Taste* Focus, or *Health* Focus. During *Natural* trials, participants were told to allow whatever feelings and thoughts came most naturally to mind and to choose according to their preferences on that trial. During *Taste* trials, participants were asked to focus on how tasty each food was and try to bring their actions in line with this consideration. During *Health* trials, participants were asked to focus on the health implications of their choice. As in the Altruistic Choice Task, attentional instructions were given prior to each block of 10 trials, and participants were asked to focus on the specific instruction given for all trials within a block. However, participants knew that they would receive the outcome of one of their choices and were told that they should choose according to their preferences regardless of the instruction, thus encouraging participants to choose in a way that reflected their current decision value for the item.

# Defining normative choice

## Behavioral definition of generosity

All choices involved a tradeoff between maximizing outcomes for the self or the other. We therefore label specific decisions as normative (i.e., generous) if the participant accepted a proposal when $Self < $Other or rejected one when $Self > $Other. Choices were labeled as hedonistic (i.e., selfish) otherwise.

## Behavioral definition of healthy choice

In the Dietary Choice Task, we separately examined trials requiring a tradeoff between taste and health (i.e., conflict trials where a food was rated either as healthy but not tasty, or as unhealthy but tasty) as well as trials with no tradeoff (i.e., no-conflict trials where a food was both tasty and healthy, or both unhealthy and not tasty). In both cases, we label specific decisions as normative (i.e., healthy) if the participant either accepted a healthy food or rejected an unhealthy food. All other choices were labeled as hedonistic (i.e., unhealthy).

# Computational model fitting

We used a Bayesian model-fitting approach to identify the best-fitting model parameters of the DDM (i.e., attribute-specific weight parameters W, a drift constant C, threshold B, and non-decision time ndt) to account for choices and RTs, separately for each participant in each study and each condition in Datasets 2 and 3. More specifically, we obtained estimates of the posterior distribution of each parameter using the differentially evolving Monte-Carlo Markov chain sampling method and MATLAB (*MATLAB, 2016*) code developed by *Holmes and Trueblood, 2018*. This method uses the DDM described above (computational model simulations) to simulate the likelihood of the observed data (i.e., choices and RTs) given a specific combination of parameters. We then used this likelihood to

construct a Bayesian estimate of the posterior distribution of the likelihood of the parameters given the data.

For each individual fit, we used 3 × N chains, where N is the number of free parameters (6 in Datasets 1 and 2, 5 in Dataset 3), using uninformative, uniformly distributed priors and constraining parameter values as shown in *Supplementary file 1* based on previous work (*Tusche and Hutcherson, 2018*; *Hutcherson et al., 2015*) and theoretical bounds. To construct the estimated posterior distributions of each parameter, we sampled 1500 iterations per chain after an initial burn-in period of 500 samples. Best-fitting values of each parameter were computed as the mean over the posterior distribution for that parameter. These parameter values (see *Supplementary file 1*) were used to simulate trial-by-trial neural activation for use in the GLMs described below. Importantly, parameter values identified by this fitting procedure suggested that the model provided a good fit to behavior across all three datasets (*Figure 4—figure supplement 1*). The analysis of the values of individual chains using the Gelman-Rubin statistic (*Gelman and Rubin, 1992*) indicated good convergence to the posterior distribution for each parameter (all R < 1.05).

## Neuroimaging analyses

*GLM 1a: Neural Correlates of the DDM (Dataset 1)*. We used GLM 1a to identify brain regions where activation varied parametrically according to the predictions of the DDM in Dataset 1 (Altruistic Choice Task). We hypothesized that this analysis would identify brain areas such as the dlPFC commonly associated with cognitive control and provide further data-driven support for the independent dlPFC-ROI. For each trial, we determined the BOLD approximation of the DDM as the value consisting of the sum total of the simulated neural response, averaged over all simulations terminating in the observed choice on that trial within ±250 ms of the observed RT. This estimate was then used as a parametric regressor, modulating a boxcar function with onset at the beginning of the choice period and having a duration of the RT on that trial (see Appendix 1 for further detail on selecting the best regressor). To simulate expected DDM activation on each trial, we generated 10,000 simulations using the best-fitting parameters for each participant and the estimated value of the proposal and default on each trial (i.e., $w_{Self}$*\$Self + $w_{Other}$*\$Other + $w_{Fairness}$*|\$Self − \$Other|+C).

Then, for each subject, we estimated a GLM with AR(1) and the following regressors of interest: (R1) A boxcar function for the choice period on all trials (duration = RT on that trial). (R2) R1 modulated by the subject's stated preference on that trial (1=Strong No, 4=Strong Yes). (R3) R1 modulated by the estimated activation of the DDM on that trial. (R4) A boxcar function of 3 s specifying the outcome period on each trial. (R5) R4 modulated by the outcome for the self on each trial. (R6) R4 modulated by the outcome for the partner on each trial. (R7) A boxcar function (duration = 4 s) specifying missed trials. Parametric modulators were orthogonalized to each other in SPM (https://www.fil.ion.ucl.ac.uk/spm/). Regressors of non-interest included six motion regressors as well as session constants.

We then computed subject-level contrasts of the DDM parametric modulator (R3) against an implicit baseline. Finally, to test the hypothesis that DDM responses might correlate with activation in the dlPFC at the group level, we subjected this contrast to a one-sample t-test against zero (as implemented in SPM), thresholded at a whole-brain cluster-corrected value of p<0.05, using a cluster-defining threshold of p<0.001.

## GLM 1b: Neural correlates of the DDM (Dataset 2)

GLM1b was similar to GLM1a, except that we estimated regressors for each goal condition separately. R1, R4, and R7 were boxcar functions representing the choice period for the *Natural*, *Ethics*, and *Partner* conditions, respectively. R2, R5, and R8 modulated R1, R4, and R7 with the decision value on that trial. R3, R6, and R9 modulated R1, R4, and R7 using the estimated activation of the DDM on that trial. A single contrast representing neural correlates of the DDM was constructed by combining R3, R6, and R9 at the subject-level and performing a one-sample t-test against zero. We report results, thresholded at a voxel-wise p<0.001 and a whole-brain cluster-corrected level of p<0.05.

## GLM1c: Neural correlates of the DDM (Dataset 3)

GLM1c was similar to GLM1b but applied to the Food Choice Task in Studies 3a–c. R1, R4, and R7 were boxcar functions representing *Natural*, *Taste*, and *Health* focus conditions. R2, R5, and R8 were parametric modulators representing the decision value on that trial, and R3, R6, and R9 were

modulators consisting of the estimated DDM activity on that trial. Correlates of the DDM were identified in this study thresholded at a voxel-wise p<0.001 and a whole-brain cluster-corrected level of p<0.05.

*GLM 2a: Generous vs. Selfish decisions in Altruistic Choice (Dataset 1).* We used GLM 2a to test predictions about neural activation on trials in which subjects chose generously or selfishly. The analysis was carried out in two steps.

First, for each subject, we estimated a GLM with AR(1) and the following regressors of interest: (R1) A boxcar function for the choice period on trials when the subject chose selfishly. (R2) R1 modulated by the decision value of their 4-point preference response (i.e., Strong No to Strong Yes) at the time of choice. (R3) A boxcar function for the choice period on trials when the subject chose generously. (R4) R3 modulated by the decision value on that trial. Regressors of non-interest included six motion regressors as well as session constants.

Second, we computed the subject-level contrast image [R3 – R1], which identified brain regions with differential responses for generous compared to selfish choices. Seven subjects were excluded from this analysis for having fewer than 4 generous choices over the 180 trials. To test predictions of the DDM in regions previously associated with self-control, we conducted analyses of this contrast [Generous – Selfish] in several independently selected ROIs (see *Independently defined ROIs* section below). Independent ROIs helped avoid circular analysis and 'double-dipping' (*Kriegeskorte et al., 2009*). We also report the results of these contrasts for two other brain areas (dACC, IFG/anIns) identified as correlating with predictions of the DDM at the whole-brain level across all three datasets (see Appendix 4 and Figure 4—figure supplements 5–6).

Finally, as a supplementary analysis, we examined if any voxels beyond the ROI regions demonstrated a significant effect in any condition, using a whole-brain analysis thresholded at p<0.001, uncorrected (see *Supplementary file 4*).

*GLM 2b: Generous vs. Selfish decisions in Altruistic Choice (Dataset 2).* We used GLM 2b to test predictions about activation on trials in which the subject chose generously or selfishly in Dataset 2, and to compare how instructed attention altered these responses. GLM2b was similar to GLM2a, except that we estimated regressors for each goal condition separately. All unreported details match GLM2a. Regressors of interest consisted of the following: (R1) A boxcar function for the choice period on trials when the subject chose selfishly in *Natural Focus* trials. (R2) R1 modulated by the decision value of 4-point preference response (i.e., Strong No to Strong Yes) expressed at the time of choice. (R3) A boxcar function for the choice period on trials when the subject chose generously in *Natural Focus* trials. (R4) R3 modulated by the decision value. (R5–8) Analogous regressors for generous and selfish choices during *Ethics Focus* trials. (R9–12) Analogous regressors for generous and selfish choices during *Partner Focus* trials. (R13–15) A boxcar function of 3 s duration signaling the outcome period for *Natural, Ethics,* or *Partner Focus* trials. (R16–18) R13–15 modulated by the amount received by the subject at the outcome phase for *Natural, Ethics,* or *Partner Focus* trials. (R19–21) R13–15 modulated by the amount received by the partner at the outcome phase for *Natural, Ethics,* or *Partner Focus* trials.

We then computed the subject-level contrast images [R3 – R1], [R7 – R5], and [R11 – R9], which identified regions with differential responses for generous compared to selfish choices in each goal condition. We computed the average value of each of these contrasts within the same ROIs specified above for GLM2a. As a supplementary analysis, we also asked whether any voxels beyond these regions demonstrated a significant effect in any condition, using a whole-brain analysis thresholded at p<0.001, uncorrected (see *Supplementary file 4*).

*GLM 2c: Healthy vs. Unhealthy decisions in the Food Choice Task (Dataset 3).* GLM 2c was analogous to GLM 2b but examined healthy vs. unhealthy choices in the Dietary Choice Task, separately for conflicted trials (i.e., healthy but not tasty foods, or tasty but unhealthy foods) and for unconflicted trials (i.e., healthy and tasty foods, or unhealthy and not tasty foods). It included the following regressors of interest: (R1) A boxcar function for the choice period when the subject made a healthy choice in *Natural Focus* trials with conflict. (R2) R1 modulated by the decision value of behaviorally expressed preference at the time of choice. (R3) A boxcar function for the choice period when the subject made an unhealthy choice in *Natural* trials with conflict. (R4) R3 modulated by the decision value on that trial. (R5–8) Analogous regressors for healthy and unhealthy choices during conflicted *Taste Focus* trials. (R9–12) Analogous regressors for healthy and unhealthy choices during conflicted *Health Focus*

trials. (R13) Healthy choices on unconflicted *Natural Focus* trials. (R14) Unhealthy choices on unconflicted *Natural Focus* trials. (R15–16) R13 and R14 modulated by preference. (R17–20) Analogous regressors for healthy and unhealthy choice on unconflicted *Health Focus* trials. (R21–24) Analogous regressors for healthy and unhealthy choice on unconflicted *Taste Focus* trials. Subject-level contrast images of healthy vs. unhealthy choices, in each condition separately and separately for conflicted vs. unconflicted trials, were computed in a manner identical to GLM2b. We analyzed activation for these contrasts specifically within the independently defined ROIs described below. As a supplementary analysis, we also report results at the whole-brain level at p<0.001, uncorrected, in *Supplementary file 4*. Unreported details are as in GLM 2a.

## Independently defined ROIs

Our primary hypothesis is that regions previously associated with self-control might actually be performing computations related to post-valuation evidence accumulation, rather than pre-valuation modulation of attribute weights. Although our DDM neural results at the whole-brain level confirmed that activation in the dlPFC correlated with the DDM, we sought to avoid circularity by testing our primary predictions about dlPFC activation in ROIs defined independently of the main contrast. Toward this end, we tested activation in three ROIs. As our primary ROI, we created a left dlPFC ROI by defining a region associated with the term 'inhibitory control' in the meta-analytic database Neuro-Query, thresholded at Z=5 (see *Figure 4*, top panel, blue region). To examine the specificity of our dlPFC results for the right vs. left hemisphere, in a supplemental analysis (see Appendix 3), we flipped our dlPFC-ROI over the left-right axis (creating a second, bilaterally matched ROI). Finally, to provide further evidence for the robustness of our main results, we used a third independent-identified left dlPFC-ROI functionally defined by the contrast of healthy vs. unhealthy choice in a seminal fMRI study of dieters (*Hare et al., 2009*).

# Acknowledgements

These studies were made possibly by grants from the Gordon and Betty Moore Foundation, the Lipper Foundation, the Economic Research Service of the US Department of Agriculture on Behavioral Health Economics Research on Dietary Choice and Obesity, an Agence Nationale de la Recherche Sorbonne Universite´s Emergence Grant, and a National Institute of Mental Health Silvio O Conte award (NIMH Conte Center 2P50 MH094258). The scientific results and conclusions reflect the authors' opinions and not the views of the granting entities. We gratefully acknowledge Antonio Rangel and Todd Hare for comments on earlier versions of this manuscript.

# Additional information

## Funding

| Funder | Grant reference number | Author |
| --- | --- | --- |
| National Institute of Mental Health | NIMH Conte Center 2P50 MH094258 | Cendri A Hutcherson |

The funders had no role in study design, data collection and interpretation, or the decision to submit the work for publication.

## Author contributions

Cendri A Hutcherson, Conceptualization, Data curation, Software, Formal analysis, Funding acquisition, Investigation, Visualization, Methodology, Writing – original draft, Project administration, Writing – review and editing; Anita Tusche, Conceptualization, Data curation, Validation, Investigation, Visualization, Methodology, Writing – original draft, Project administration, Writing – review and editing

## Author ORCIDs

Cendri A Hutcherson http://orcid.org/0000-0002-4441-4809
Anita Tusche http://orcid.org/0000-0003-4180-8447

### Ethics

Human subjects: All human subjects provided informed consent, including the consent to publish. All procedures were approved by the Internal Review Board at the California Institute of Technology or the Comité; de Protection des Personnes, Ile-de-France VI, INSERM, approval #C07-28, DGS approval #2007-0569, ID-RCB approval #2007-A01125-48CPP.

### Decision letter and Author response

Decision letter https://doi.org/10.7554/eLife.65661.sa1
Author response https://doi.org/10.7554/eLife.65661.sa2

---

## Additional files

### Supplementary files

• Supplementary file 1. Table S1. Estimated Model Parameters: All Datasets. Parameter values were estimated using a differential-evolution Markov chain Monte Carlo method developed by *Holmes and Trueblood, 2018*. Parameters beginning with w indicate weighting parameters applied to different attributes. Datasets 1 and 2: proposed payoff to self vs. the default, proposed payoff to other vs. the default, fairness [|\$Self − \$Other|], and a constant bias toward the proposal; Dataset 3: tastiness and healthiness vs. the default as well as a constant bias toward the displayed food. B: choice-defining threshold. ndt: non-decision time. A priori constraints on the parameters, determined based on previous work and on theoretical limits, restricted them to the range indicated. In Datasets 2 and 3, columns indicated by different subscripts (a–c) differ significantly from each other at p<0.05, corrected for multiple comparisons. * Average parameter values collapsing over three independently collected samples of participants (see *Supplementary file 2* for details of each sample separately).

• Supplementary file 2. Table S2. Estimated Model Parameters: Dataset 3, separately by Samples 3a, 3b, and 3c. Parameter values were estimated using a differential-evolution Markov chain Monte Carlo method developed by *Holmes and Trueblood, 2018*. Parameters beginning with w indicate weighting parameters applied to tastiness and healthiness vs. the default as well as a constant bias toward the displayed food. B: choice-defining threshold. ndt: non-decision time. A priori constraints on the parameters, determined based on previous work and on theoretical limits, restricted them to the range indicated. Columns indicated by different subscripts (a–c) differ significantly from each other at p < 0.05, corrected for multiple comparisons.

• Supplementary file 3. Table S3. Neural correlates of the drift diffusion model (DDM) across datasets. Regions are reported at a voxel level of p < 0.001, uncorrected and a whole-brain FWE cluster-corrected level of p < 0.05, unless otherwise noted. * Distinct peak within larger cluster. † Significant at p < 0.005, uncorrected, reported for completeness.

• Supplementary file 4. Table S4. Neural correlates of normative vs. hedonistic choice across datasets. Regions are reported at a voxel-level threshold of p < 0.001, uncorrected, and a minimum volume of k = 10 voxels, unless otherwise noted. * Significant at p < 0.005, uncorrected and minimum volume of k=20 voxels, reported for completeness. ** Distinct peak within larger cluster, reported for completeness.

• Transparent reporting form

### Data availability

Analysis code, ROI masks, and behavioral data for all studies, as well as functional imaging data for Dataset 1, is available on the Open Science Framework at https://osf.io/eqvwd/. Functional imaging analysis data for Datasets 2 and 3a is available on the Open Science Framework at https://osf.io/wa4cs/. Functional imaging data for Dataset 3b is available at https://osf.io/3rne9/. Because it is part of a larger on-going study examining neural correlates of obesity, functional imaging data for Dataset 3c is not yet publicly available. However, requests for access to raw neuroimaging data can be made by contacting Dr. Hilke Plassmann (https://www.insead.edu/faculty-research/faculty/hilke-plassmann). All pre-processed ROI data and computational modeling data used for analysis and to create figures is available on the Open Science Framework at https://osf.io/eqvwd/. These data serve as source data for Figure 4 in the main text, and Appendix Figures 2-5.

The following datasets were generated:

| Author(s) | Year | Dataset title | Dataset URL | Database and Identifier |
|---|---|---|---|---|
| Tusche A, Hutcherson C | 2018 | Cognitive regulation alters social and dietary choice by changing attribute representations in domain-general and domain-specific brain circuits | https://osf.io/wa4cs/ | Open Science Framework, wa4cs |
| Hutcherson C, Tusche A | 2022 | A neurocomputational model of normative choice | https://osf.io/eqvwd/ | Open Science Framework, eqvwd |
| Hutcherson C, Tusche A, Hare T | 2022 | Data repository: Focusing attention on the health aspects of foods changes value signals in vmPFC and improves dietary choice | https://osf.io/3rne9/ | Open Science Framework, 3rne9 |

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

# Appendix 1

## Choosing an appropriate fMRI regressor for the DDM model (GLMs 1a, b, and c)

The DDM produces a dynamic accumulation signal that builds over hundreds of milliseconds. This raises a question about the appropriate way to model this signal in the hemodynamic response, which evolves more slowly over 5–10 s. To determine the appropriate regressor for GLMs 1a, b, and c, we simulated 5000 instantiations of the DDM for every subject and trial in Dataset 2, using a time step of 5 ms. For each subject, we then averaged the 5000 simulations at each time point to produce a single time course of total activity for a given set of trials. We convolved this simulated time course with the canonical form of the hemodynamic response function (HRF) to construct an expected BOLD time series given the inputs. We refer to this as the *ideal BOLD*. We then compared the shape of the ideal BOLD to two different possible instantiations within a traditional GLM analysis in SPM. Version 1 consisted of a parametric modulator of a stick function placed at the onset of the trial, consisting of the total activity in the DDM for each trial, $\sum_{t=1}^{RT} |E(t)|$. Version 2 consisted of a parametric modulator identical to Version 1, but modulating a boxcar function placed at the onset of the trial with a duration equal to RT for that trial. Each of these regressors was convolved with the canonical form of the HRF and correlated with the ideal time series to determine the one providing the closest match.

Results suggested that version 2 provided a closer match (Pearson's r ranging from 0.90 to 0.99, average = 0.95) compared to version 1 (Pearson's r ranging from 0.64 to 0.93, average = 0.82). Importantly, a parametric regressor consisting simply of the observed RT on each trial modulating a boxcar of duration equal to the RT also provided an excellent, and even slightly better, match (Pearson's r ranging from 0.93 to 0.99, average = 0.97). We thus suspect that it might be difficult in practice to distinguish between computations related solely to the DDM from those related to RT. Note also that the inclusion of the unmodulated boxcar function with a duration equal to the RT on each trial controls for non-specific activation related to RTs that does not build over time in the manner expected based on the DDM, although this unmodulated regressor is itself highly correlated with the predicted BOLD signal from the DDM (average r=0.96). Despite this high correlation, simulations suggested that both the mean-centered DDM parametric modulator and the mean-centered RT parametric modulator continue to show a significant association with the DDM-related BOLD signal even after including the unmodulated boxcar regressor in the model (both p<0.0001). Thus, we think that these two parametric modulators provide good markers of candidate regions for the evidence accumulation function.

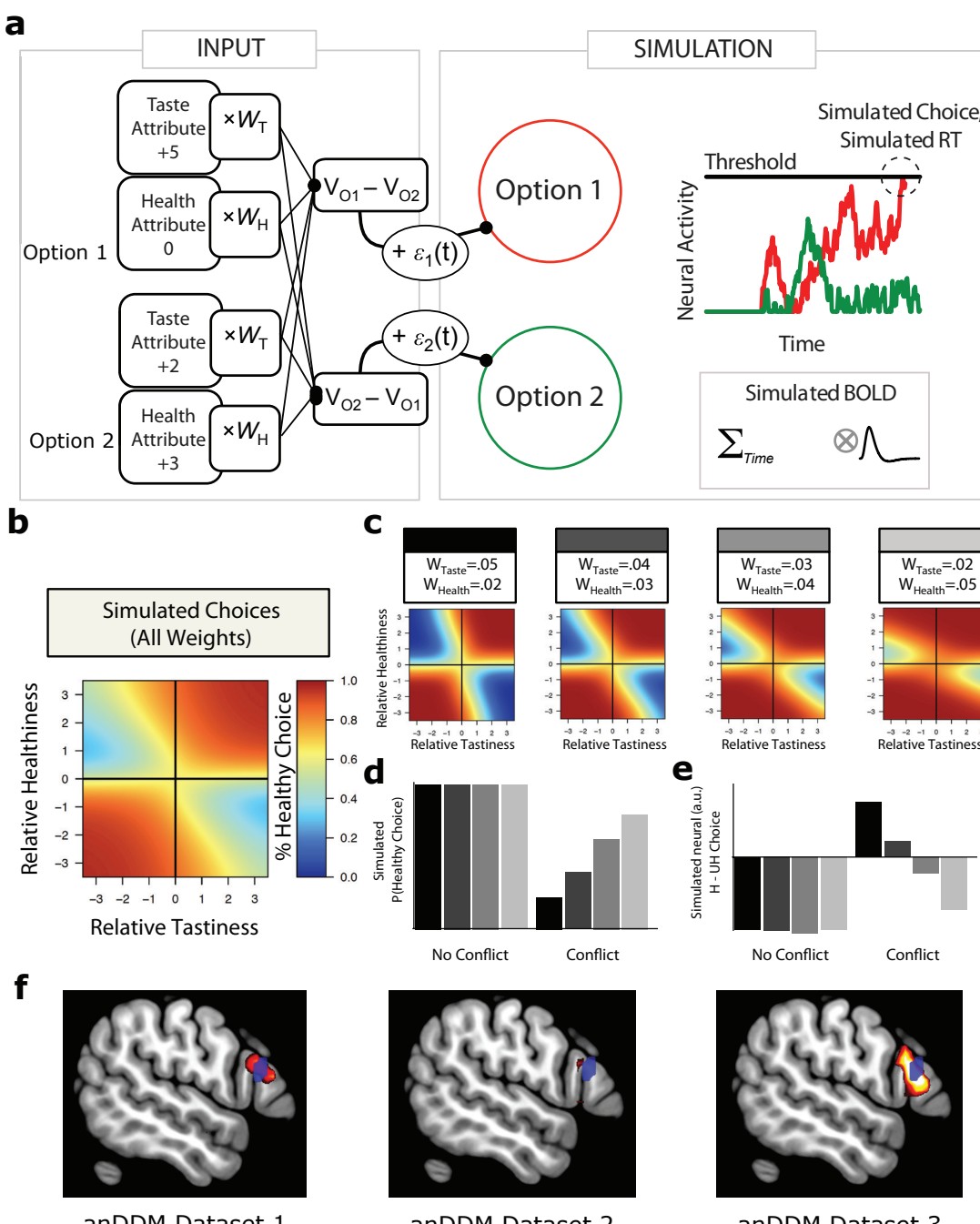

**Appendix 1—figure 1.** Simulating the dilemma of self-control using the attribute-based neural drift diffusion model (anDDM). (**a**) An evidence accumulation model (anDDM) can be used to simulate decision making for any self-control context requiring an integration of normative and hedonistic considerations (dietary choices displayed, integrating relative tastiness, and healthiness). (**b**) On average across multiple different goals, the likelihood of a healthy choice depends on the relative attribute values of one option vs. another, and is less likely when tastiness and healthiness conflict. Warmer colors indicate a higher likelihood of choosing the healthier option. (**c**) As with the DDM, specific goals prioritizing tastiness or prioritizing healthiness alter the frequency of healthy choice, although it still depends on the relative values of both tastiness and healthiness. Four example, decision makers with different weights on healthiness and tastiness are displayed for comparison. (**d**) The overall likelihood of a healthy choice (averaged separately for all combinations of conflict or no-conflict choices) depends on weights. Goals prioritizing tastiness (darker bars) produce fewer healthy choices than goals prioritizing healthiness (lighter bars), but only when tastiness and healthiness conflict. (**e**) The computational model can also simulate expected neural activity (i.e., aggregate activity in the two neuronal pools, summed over decision time: ) when choosing

*Appendix 1—figure 1 continued on next page*

*Appendix 1—figure 1 continued*

healthy [H] or unhealthy [UH] options, as a function of relative option values and different goals. Bars display the overall difference in neural activity for H compared to UH choices for goals prioritizing tastiness (darker bars) and healthiness (lighter bars), divided as a function of attribute conflict. In no-conflict trials, healthy choices elicit less activity regardless of the goal (i.e., Activity H Activity UH), but only when goals prioritize tastiness. (**f**) Neural activation in the left dlPFC match estimates of the anDDM on that trial (displayed at p < 0.001, uncorrected in left and right-most images, p = 0.005, uncorrected in middle image). Blue region of interest (ROI) indicates the left dlPFC area identified as correlating with self-control using the meta-analytic database NeuroQuery (*Dockès et al., 2020*).

## Appendix 2

## Alternative computational model specification (anDDM)

In the main paper, we used the canonical form of the DDM (*Rudorf and Hare, 2014*) to simulate choices, RTs, and neural response. However, other work( *Hare et al., 2011b*) by Hare and colleagues has proposed that accumulator activity is best described by a variant of this model, which we hereafter refer to as an anDDM. The anDDM reported on by Hare and colleagues has the advantage of being more neurobiologically plausible. However, because initial efforts to fit choices using this alternative specification of the choice process indicated relatively poorer convergence of key computational parameters (as measured by the Gelman-Rubin statistic, i.e., R-hat > 1.1 for several parameters), as well as issues in properly identifying the values of some of the parameters in this model, we opted in favor of describing results in terms of the simpler DDM in the main text. However, here, we show that results are similar for all key analyses if this more neurally plausible model is used instead.

Similar to the DDM, the anDDM assumes that the brain makes decisions through a process of value-based attribute integration and competition. However, in contrast to the DDM, which uses a single, one-dimensional evidence accumulation process, in the anDDM choices are resolved via evidence accumulation in two spatially intermingled populations of neurons representing the options under consideration (here denoted as Option 1 and Option 2), with instantaneous firing rates (FR) at time t of $FR_1(t)$ and $FR_2(t)$. At the beginning of the choice period $FR_1(0)=FR_2(0)=0$. In the version of this model reported on here, firing rates in each population evolve dynamically from the onset of choice based on the sum total of excitatory inputs (detailed below). A choice results at time t', the first moment at which the firing rate of one of the two populations exceeds a predetermined threshold or barrier B. The total RT is t' plus a constant non-decision time (ndt) that accounts for perceptual and motor delays (see *Appendix 1—figure 1a*).

Firing rates in the two pools evolve noisily over time according to the following two equations:

$$\left\{ \begin{array}{l} FR_1\left(t\right) = \max\left(0, \left(v_1 - v_2\right) + \varepsilon_1\left(t\right)\right) \\ FR_2\left(t\right) = \max\left(0, \left(v_2 - v_1\right) + \varepsilon_2\left(t\right)\right) \end{array} \right\}$$

where the noise terms $\varepsilon_x\left(t\right)$ are normally distributed ~ N(0,.1), and $v_1$ and $v_2$ represent external inputs proportional to the overall values of Options 1 and 2, determined by the weighted sum of their choice-relevant attribute values:

$$\left\{ \begin{array}{l} v_1 = \sum_i w_i Attrib_i^1 \\ v_2 = \sum_i w_i Attrib_i^2 \end{array} \right\}.$$

Thus, each pool's activity receives an external input proportional to its value relative to the other option. We also tested several variants of this basic neural model, including lateral inhibition between the two neuronal pools, recurrent excitation/inhibition of each pool onto itself, and raw rather than relative values (i.e., $v_1$ and $v_2$ rather than $v_1 - v_2$) as inputs to each neuronal pool. However, model-fitting exercises generally indicated that the simpler model, with no lateral inhibition or recurrent excitation, provided the most parsimonious fit to the data, and that the values of parameters related to lateral inhibition and recurrent excitation showed poor identifiability. We thus proceeded with simulations of neural activity using the simple model with two independently evolving accumulators.

In our simulations, we assumed that value inputs to the accumulators depend on two independent attributes: one related to hedonism (e.g., tastiness of a food) and one related to norms and standards (e.g., healthiness), although in principle, any number and type of attribute could occur. Using these equations allowed us to simulate the dynamically evolving activation across the two neuronal populations, and to derive distributions of both response times (RTs) and neural response. As with the DDM, we label the final output (i.e., choice) of the system as 'normative' if it results in selecting the option with the higher unweighted value for the normative attribute (e.g., the option with higher healthiness).

To establish comparability between the predictions of the DDM and the anDDM, we focus here on two results described in the main text using the DDM: behavioral predictions and the neural prediction that activity in the dlPFC is associated with predicted response. To establish that the anDDM made similar behavioral predictions, we engaged in a similar set of simulation exercises as

in the main text, but substituting the anDDM model described above for the DDM. As expected, we observed nearly identical predictions of the anDDM regarding the relationship between choices, RTs, neural response, and the weights on hedonistic vs. virtuous attributes. To illustrate this comparability, we recreated *Figure 2* of the main manuscript (which shows predictions of the DDM) using the anDDM (see *Appendix 1—figure 1b-e*).

Unsurprisingly, given the close correspondence of predictions, we found similar results both behaviorally and neurally when using the anDDM instead of the DDM to examine responses. When fitting the anDDM to choices, we found that weights on attributes correlated highly with weights identified using the DDM (r values ranging from 0.98 to 0.99). Furthermore, we found highly similar regions of the dlPFC in Datasets 1, 2, and 3 when using the anDDM's predicted activity instead of the DDM in GLMs that were otherwise identical to GLMs 1a–c reported in the main text (see *Appendix 1—figure 1*). Thus, our conclusions are robust to alternative model specifications, and likely represent a general principle of decision making via evidence accumulation.

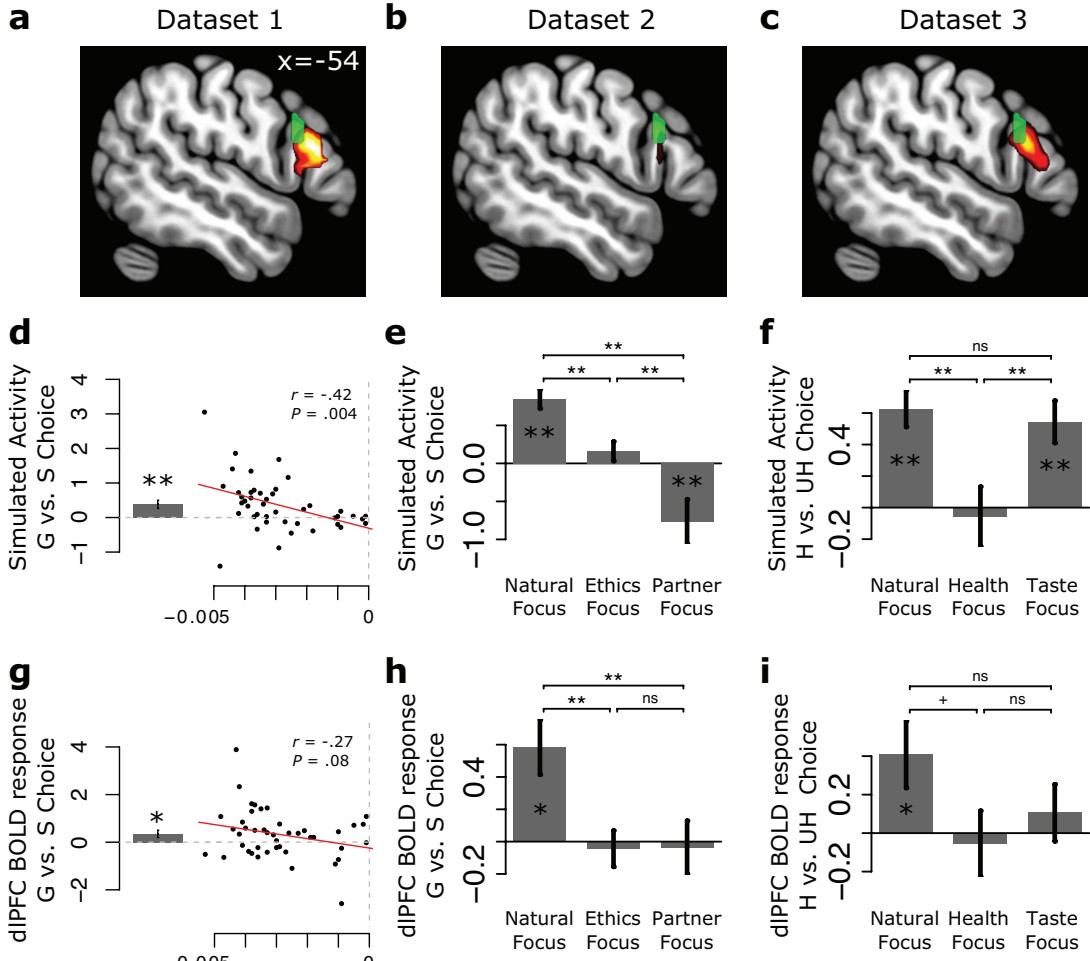

**Appendix 2—figure 1.** BOLD responses in a left dorsolateral prefrontal cortex (dlPFC) region associated with self-control in dieters. Top (dlPFC): Trial-by-trial BOLD response in the dlPFC correlates significantly with predicted activity of the drift diffusion model (DDM) across three separate datasets. This includes responses during altruistic choice (**a, b**) and dietary choice (**c**). Hot colors display DDM-related responses thresholded for display purposes at p < 0.001 (**a and c**) and p < 0.005 (**b**), uncorrected. The location of this activity was anatomically similar to an area of the dlPFC associated with self-control in dieters (*Hare et al., 2009*) (green region of interest [ROI] displayed in each panel), which served as a supplementary ROI in which to test predictions of the computational model (**d–i**). Middle (Model Simulations): Given participants' observed attribute weights, we simulated expected neural response during normative vs. hedonistic choice, using the DDM, for all trials in Dataset 1 (**d**), as well as separately for each condition in Dataset 2 (**e**) and Dataset 3 (**f**). Bottom (dlPFC Results): Within the independently defined
*Appendix 2—figure 1 continued on next page*

*Appendix 2—figure 1 continued*

dlPFC ROI associated with self-control in dieters (green voxels in a–c), BOLD response during normative choice vs. hedonistic choice when attributes conflict, in (**g**) Dataset 1 (N = 50) for all trials, as well as in (**h**) Dataset 2 (N = 49) and (**i**) Dataset 3 (N = 113) as a function of regulatory goals. Regulatory goals were designed to decrease hedonic weights and increase normative weights on choice (Dataset 2: Ethics, Partner; Dataset 3: Health) or to put weights on hedonic attributes (Dataset 3: Taste). Only conflict trials are considered across datasets. As predicted, normative choices activate the dlPFC, but only when goals result in a greater weight on hedonistic than normative attributes. Contexts that increase normative attribute weights yield less dlPFC activity during normative choices (Dataset 2: Ethics, Partner; Dataset 3: Health). *p < 0.05 based on one-sample t-tests against 0, or paired-tests between conditions; **p < 0.01. Error bars show standard error of the mean.

# Appendix 3

## Alternative ROI specifications

### Left dlPFC region associated with self-control in dieters

In the main paper, we tested predictions of the DDM for the comparison of normative vs. hedonistic choice across our three datasets using an independent ROI based on a NeuroQuery (*Lopez et al., 2014*) meta-analysis of inhibitory control. This ROI has the advantage of being based on activation across hundreds of studies and thousands of participants covering a variety of domains, as well as being linguistically associated with one of the key terms of interest from self-control (i.e., inhibitory control). However, alternative specifications of the dlPFC are possible. Here, we report the results of a second, alternative ROI, based on the results of a comparison of healthy vs. unhealthy food choices (thresholded at a voxel-level p=0.001, see *Appendix 2—figure 1*, green ROI) in a seminal study examining self-control success vs. failure (i.e., healthy vs. unhealthy choice) in dieters (N=37[1]). (We thank Todd Hare for generously sharing a mask of this ROI with us.)

In Dataset 1 (unregulated altruistic choice), we replicated the result from the main manuscript that the dlPFC is more activated on generous compared to selfish choices ($t_{43}$=2.09, p=0.04, *Appendix 2—figure 1g*), and similarly observed a marginally significant correlation with difference in weights on self vs. other (r=–0.27, p=0.08).

In Dataset 2, we replicated our observation that condition (Natural, Ethics, Partner) had a significant effect on the contrast of generous vs. selfish choices in this ROI ($F_{2,96}$ = 6.63, p=0.002, *Appendix 2—figure 1h*), and confirmed that this was due to a significant difference in generous vs. selfish choices in the Natural condition (paired-$t_{48}$=3.42, p=0.001) that disappeared in each of the other two conditions (G vs. S choice in Ethics vs. Natural paired-$t_{48}$=3.59, p=0.001; G vs. S choice in Partner vs. Natural paired-$t_{48}$=2.93, p=0.001).

Finally, we tested predictions in Dataset 3. While the repeated-measures of ANOVA of the effect of condition on healthy vs. unhealthy choice fell short of significance ($F_{2,177}$ = 2.32, p=0.099, *Appendix 2—figure 1i*), planned post hoc comparisons showed similar effects to the ROI reported in the main paper. Healthy choices were associated with significantly greater activity than unhealthy choices in the Natural condition (paired-$t_{100}$=2.34, p=0.02). However, this difference disappeared in the Health-focused condition (paired-$t_{100}$=0.32, p=0.75), a drop that differed significantly from Natural trials (H vs. UH choice in Health vs. Natural trials paired-$t_{93}$=1.98, p=0.05). As with the meta-analytic ROI reported in the main paper, the contrast of healthy vs. unhealthy choices in the Taste condition was positive but non-significant, lay in between Natural and Health conditions, and did not differ significantly from either (both p>0.27).

## Right dlPFC equivalent of the left dlPFC NeuroQuery ROI

Although our results have so far focused on left-hemisphere ROIs, we also sought to determine whether our results were comparable in the right hemisphere. Toward this end, we mirrored the NeuroQuery ROI used in the main manuscript over the midline, and performed a similar set of analyses of normative vs. hedonistic choice within each dataset. Results largely replicated findings in the left dlPFC (see *Appendix 3—figure 1*).

In Dataset 1, we find a significant difference between generous and selfish choices in the right dlPFC (paired-$t_{43}$=4.56, p=0.00004). This effect is actually stronger than the corresponding effect in the left dlPFC (paired-$t_{43}$ of effect in right vs. left dlPFC = 4.15, p=0.0001). We similarly observed a marginally significant correlation with difference in weights on self vs. other (Pearson's r=–0.27, p=0.08).

In Dataset 2, we replicated all of our main findings in the right dlPFC. In Natural trials, generous choices elicited significantly more activity than selfish choices (paired-$t_{48}$=2.20, p=0.03). We also observe a significant effect of condition on this difference ($F_{2,96}$ = 5.14, p=0.008). Unpacking this, we observed that generous compared to selfish choices evoked non-significantly higher activity in the Ethics Focus condition, a difference that is marginally different from Natural trials (paired-$t_{48}$=1.81, p=0.04, one-tailed). In contrast, generous decisions in the Partner Focus condition evoke non-significantly *lower* activity compared to selfish decisions (paired-$t_{48}$=1.37, p=0.18), a difference which is significantly lower than Natural trials (paired-$t_{48}$=2.89, p=0.006).

In Dataset 3, we find similar though somewhat weaker results in the right dlPFC than in the left dlPFC. Specifically, we find that in the right dlPFC, healthy choices elicit significantly greater

activity than unhealthy choices in Natural trials (paired-$t_{100}$=1.79, p=0.04, one-tailed). We also find a marginally significant effect of condition on this effect ($F_{2,181}$ = 2.63, p=0.07), driven by a significantly reduced difference between healthy and unhealthy choices in the Health Focus condition compared to both Natural trials (difference of differences $t_{93}$=1.86, p=0.03, one-tailed) and, unexpectedly, to Taste trials ($t_{83}$=1.69, p=0.05, one-tailed).

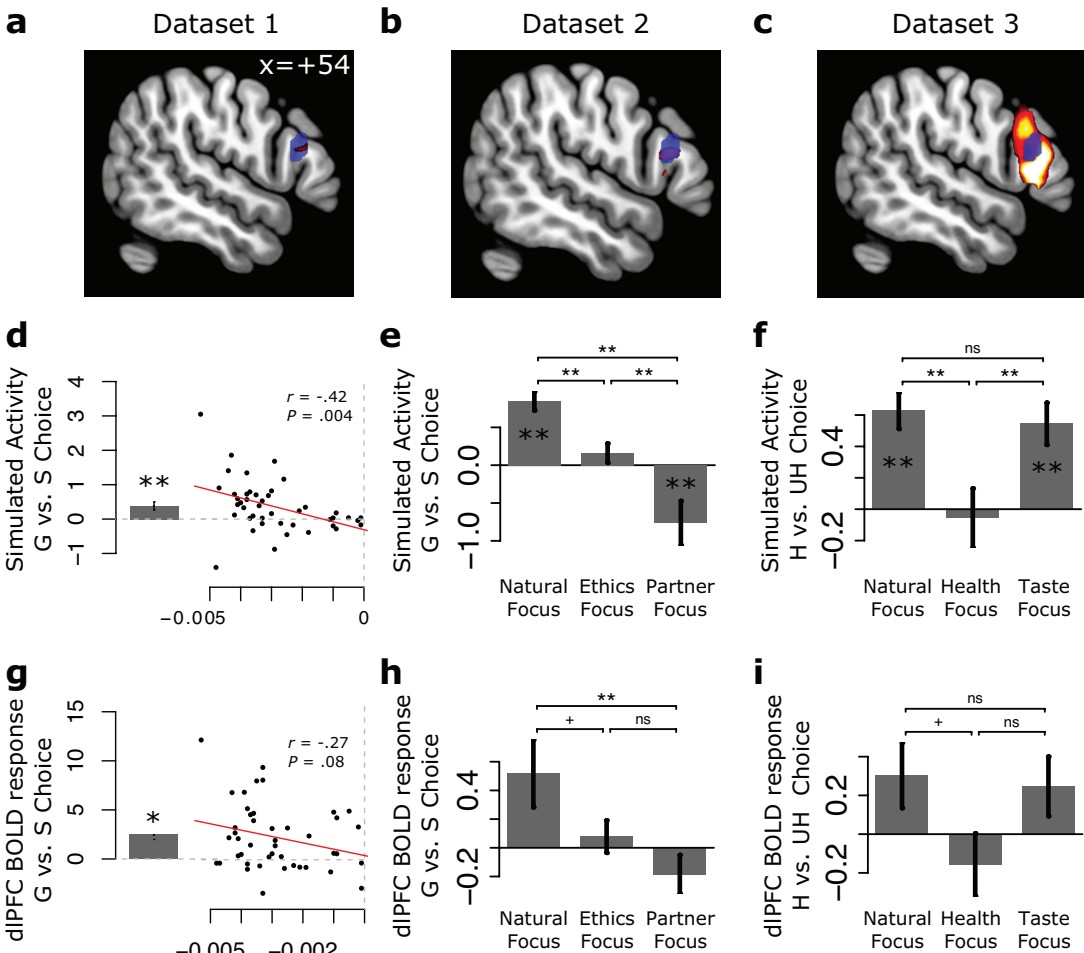

**Appendix 3—figure 1.** BOLD responses in the right dorsolateral prefrontal cortex (dlPFC) during self-control dilemmas. Top (dlPFC): Trial-by-trial BOLD response in the right dlPFC correlates significantly with predicted activity of the drift diffusion model (DDM) across three separate datasets. This includes responses during altruistic choice (**a, b**) and dietary choice (**c**). Hot colors display DDM-related responses thresholded for display purposes at p < 0.001, uncorrected. The location of this activity is anatomically similar to an area of the left dlPFC, mirrored over the x-axis to the right hemisphere, and associated with inhibitory control in a meta-analysis *Dockès et al., 2020* (blue ROI displayed in each panel), which served as a supplementary region of interest (ROI) in which to test predictions of the computational model (**d–i**). Middle (Model Simulations): Given participants' observed attribute weights, we simulated expected neural response during normative vs. hedonistic choice, using the DDM, for all trials in Dataset 1 (d), as well as separately for each condition in Dataset 2 (**e**) and Dataset 3 (**f**). Bottom (dlPFC Results): Within the independently defined right dlPFC ROI (blue voxels in **a–c**), BOLD response during normative choice vs. hedonistic choice when attributes conflict, in (**g**) Dataset 1 (N = 50) for all trials, as well as in (**h**) Dataset 2 (N = 49) and (**i**) Dataset 3 (N = 113) as a function of regulatory goals. Regulatory goals were designed to decrease hedonic weights and increase normative weights on choice (Dataset 2: Ethics, Partner; Dataset 3: Health) or to put weights on hedonic attributes (Dataset 3: Taste). Only conflict trials are considered across datasets. As predicted, normative choices activate the dlPFC, but only when goals result in a greater weight on hedonistic than normative attributes. Contexts that increase normative attribute weights yield less dlPFC activity during normative choices (Dataset 2: Ethics, Partner; Dataset 3: Health). *p < 0.05 based on one-sample t-tests against 0, or paired-tests between conditions; **p < 0.01. Error bars show standard error of the mean.

# Appendix 4

## Additional regions correlating with the DDM

In the main paper, we focus on the effects of normative vs. hedonistic choice within the independent dlPFC ROI defined by a region associated with the term 'inhibitory control' in the meta-analytic database NeuroQuery. We also ran supplemental whole-brain analyses to identify other brain areas whose activity correlated with the DDM across all three datasets. In addition to the dlPFC, we identified two other regions in this data-driven analysis approach: the dACC (see *Appendix 4—figure 1*) and right IFG/aIns (see *Appendix 4—figure 2*) whose activity correlated with the DDM across all three datasets (p<0.001, FWE cluster corrected at p<0.05 within each of the three datasets). Here, we report analogous results on measures of BOLD response in these regions during normative vs. hedonistic choice, for the sake of completeness. Note that, these key tests of our theory are technically independent of the ROI definition (i.e., ROIs are defined by parametric correlation with evidence accumulation across multiple contexts and studies, not the contrast between normative and hedonistic choices). However, we acknowledge that there is conceptual overlap, and thus some concern about analytical circularity. We thus present these results for illustrative purposes only to supplement and extend the findings we observe in our fully independent dlPFC ROI. As can be see below, these supplemental results suggest that our dlPFC findings – which we focus on in the main text – are a general principle of areas correlating with DDM response.

## dACC response during normative vs. hedonistic choices in Datasets 1, 2, and 3

We began by examining whether activity in the dACC correlated with the contrast of normative (generous) vs. hedonistic (selfish) choices in Dataset 1. As expected, and similar to the dlPFC, this region showed a significantly greater response during generous compared to selfish choices (paired-$t_{43}$=3.423, p=0.001, *Appendix 4—figure 1g*), and a negative correlation between the asymmetry of other vs. self weights and observed differences in generous vs. selfish choice (r=–0.37, p=0.01). Similarly, in Dataset 2, we observed a significant effect of normative goals on the difference in response between normative and hedonistic choices ($F_{2,96}$ = 14.03, p=14.51 × 10$^{-6}$). Follow-up t-tests confirmed that this was driven by a stronger response in the dACC to normative (generous) choices in Natural trials (paired-$t_{48}$=3.67, p=0.0006) as well as significantly stronger response to *hedonistic* choices (paired-$t_{48}$=2.3, p=0.03, difference from Natural paired-$t_{48}$=4.46, p=4.92 × 10$^{-5}$) during Partner-focused trials (*Appendix 4—figure 1h*). Finally, we replicated a similar pattern of effects in Dataset 3, showing a significant influence of normative (i.e., health-focused) goals on the contrast of normative vs. hedonistic choices ($F_{2,181}$ = 3.23, p=0.04), which was driven by a marginally stronger response on normative (healthy) choices in the Natural condition ($t_{100}$=1.84, p=0.04, one-tailed), and a marginally stronger response on *hedonistic* (i.e., unhealthy) choices during Health Focus trials (paired-$t_{100}$=1.73, p=0.04, one-tailed) (*Appendix 4—figure 1i*).

## lIFG response during normative vs. hedonistic choices in Datasets 1, 2, and 3

We next examined whether activity in the left IFG correlated with the contrast of normative (generous) vs. hedonistic (selfish) choices in Dataset 1. As expected, and similar to both the dlPFC and the dACC, this region showed a significantly greater response during generous compared to selfish choices (paired-$t_{43}$=3.14, p=0.003, *Appendix 4—figure 2g*), and a negative correlation between the asymmetry of other vs. self weights and observed differences in generous vs. selfish choice (r=–0.33, p=0.03). Similarly, in Dataset 2, we observed a significant effect of normative goals on the difference in response between normative and hedonistic choices ($F_{2,96}$ = 22.46, p=2.56 × 10$^{-7}$). Follow-up t-tests confirmed that this was driven by a stronger response in the IFG to normative (generous) choices in Natural trials (paired-$t_{48}$=4.71, p=2.155 × 10$^{-5}$) as well as significantly stronger response to *hedonistic* choices (paired-$t_{48}$=2.78, p=0.008, difference from Natural paired-$t_{48}$=5.1, p=5.717 × 10$^{-6}$) during Partner-focused trials (*Appendix 4—figure 2h*). Finally, we replicated a similar pattern of effects in Dataset 3, showing a significant influence of normative (i.e., health-focused) goals on the contrast of normative vs. hedonistic choices ($F_{2,181}$ = 3.42, p=0.03), which was driven by a significantly stronger response on normative (healthy) choices in the Natural condition ($t_{100}$=2.89, p=0.005), and a non-significantly stronger response on *hedonistic* (i.e., unhealthy) choices during Health Focus trials (paired-$t_{100}$=1.07, p=0.29, difference between Health and Natural paired-$t_{93}$=2.63, p=0.01) (*Appendix 4—figure 2i*).

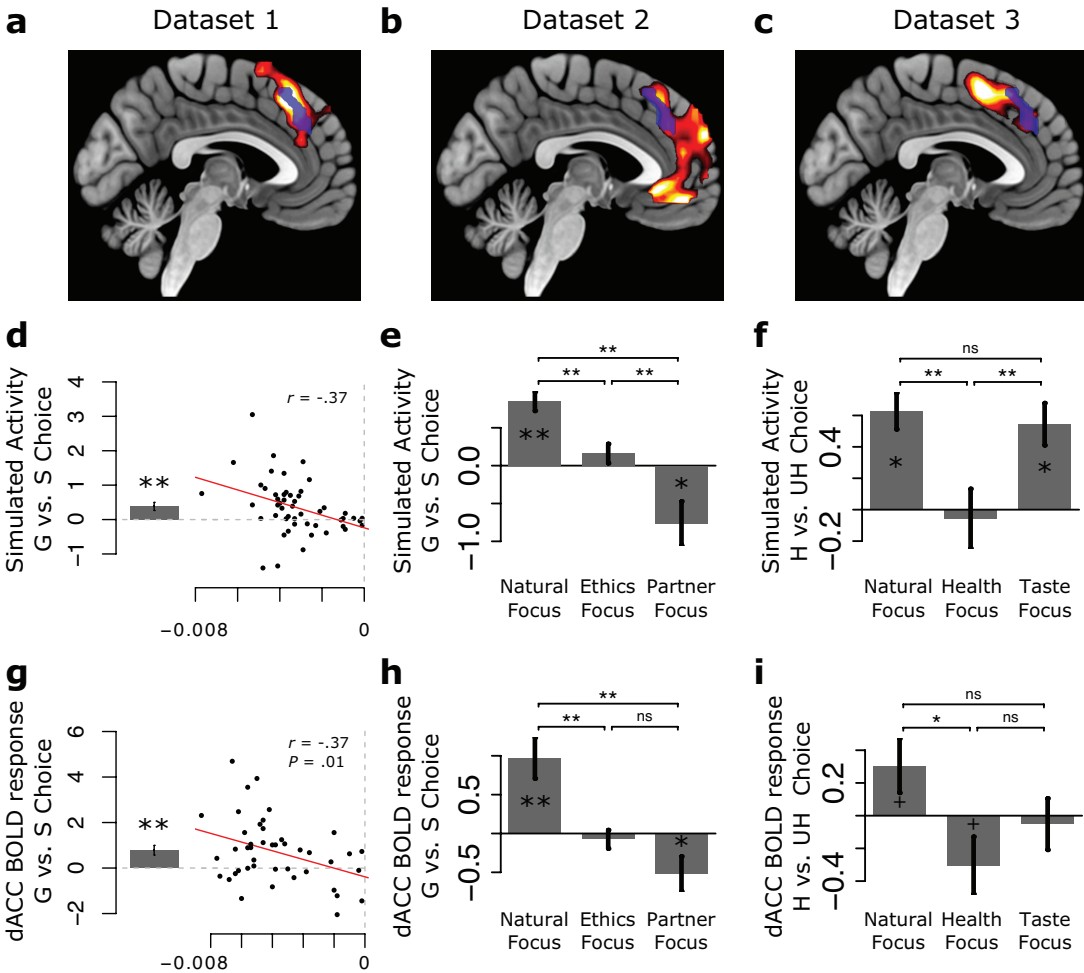

**Appendix 4—figure 1.** BOLD responses in the dorsal anterior cingulate cortex (dACC) during self-control dilemmas. Top (dACC): Trial-by-trial BOLD response in the dACC correlates significantly with predicted activity of the drift diffusion model (DDM) across three separate datasets (hot colors display DDM-related responses thresholded at p < 0.001, uncorrected for display purposes). This includes responses during altruistic choice (**a, b**) and dietary choice (**c**). Voxels of overlap across all three datasets are indicated in blue, and served as a region of interest (ROI) to extract BOLD responses during normative and hedonistic choices in each dataset (**g–i**). Here, we show for illustrative purposes only that activity in this same ROI shows similar patterns of response on normative vs. hedonistic choices as the left dlPFC and resembles model predictions (**d–f**). Middle (Model Simulations): Given participants' observed attribute weights, we simulated expected neural response during normative vs. hedonistic choice, using the DDM, for all trials in Dataset 1 (**d**), as well as separately for each condition in Dataset 2 (**e**) and Dataset 3 (**f**). Bottom (dACC Results): Within the dACC ROI associated with the DDM (blue voxels in a–c), BOLD response differed as predicted during normative choice vs. hedonistic choice when attributes conflict, in (**g**) Dataset 1 (N = 50) for all trials, as well as in (**h**) Dataset 2 (N = 49) and (**i**) Dataset 3 (N = 113) as a function of regulatory goals. Regulatory goals were designed to decrease hedonic weights and increase normative weights on choice (Dataset 2: Ethics, Partner; Dataset 3: Health) or to put weights on hedonic attributes (Dataset 3: Taste). Only conflict trials are considered across datasets. As predicted, normative choices activate the dACC, but only when goals result in a greater weight on hedonistic than normative attributes. Contexts that increase normative attribute weights yield less dACC activity during normative choices (Dataset 2: Ethics, Partner; Dataset 3: Health). +p < 0.05, one-tailed; *p < 0.05, two-tailed; **p < 0.01, two-tailed, based on one-sample t-tests against 0, or paired-tests between conditions. Error bars show standard error of the mean

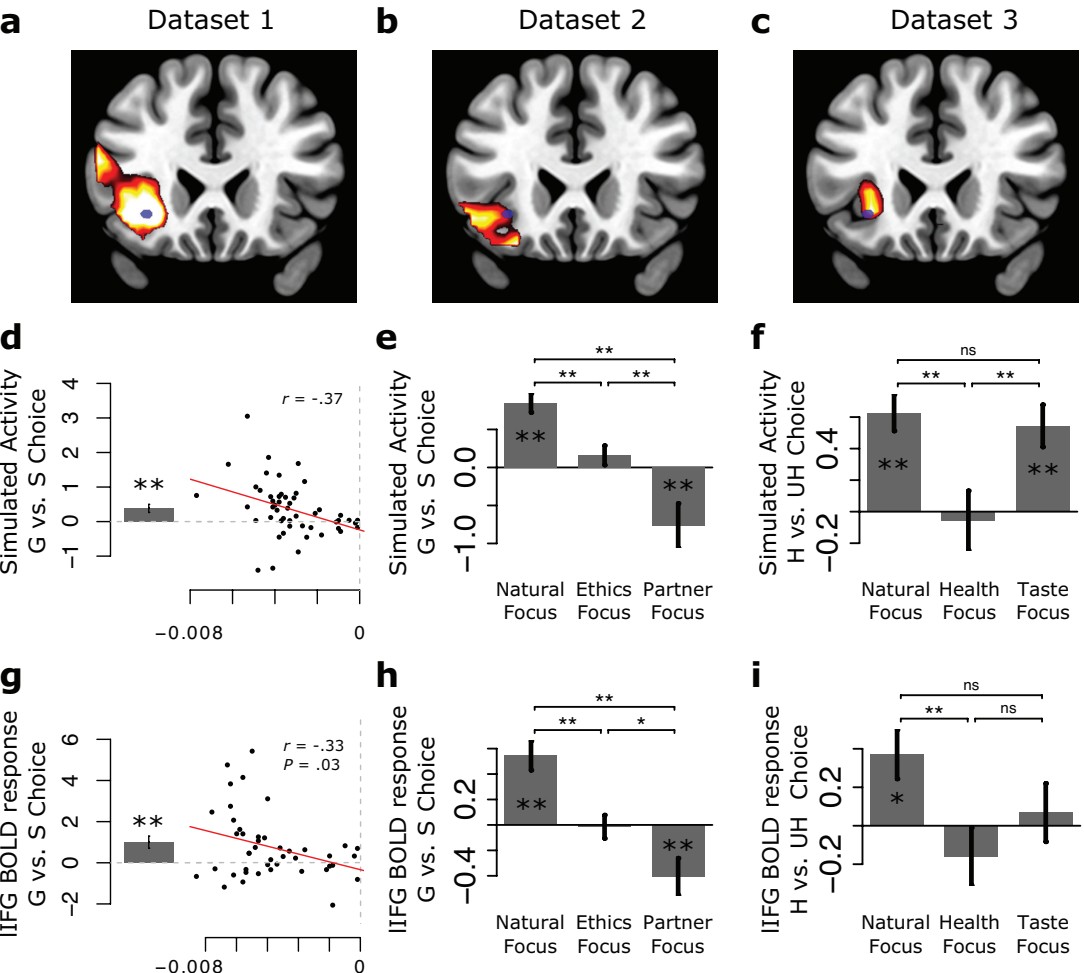

**Appendix 4—figure 2.** BOLD responses in the inferior frontal gyrus/anterior insula (IFG/aIns) during self-control dilemmas. Top (lIFG/anterior insula [aIns]): Trial-by-trial BOLD response in the lIFG/aIns correlates significantly with predicted activity of the drift diffusion model (DDM) across three separate datasets (hot colors display DDM-related responses thresholded at Pp < 0.001, uncorrected for display purposes). This includes responses during altruistic choice (**a, b**) and dietary choice (**c**). Voxels of overlap across all three datasets are indicated in blue, and served as an region of interest (ROI) to extract BOLD responses during normative and hedonistic choices in each dataset (**g–i**). Here, we show for illustrative purposes only that activity in this same ROI shows similar patterns of response on normative vs. hedonistic choices as the left dlPFC and resembles model predictions (**d–f**). Middle (Model Simulations): Given participants' observed attribute weights, we simulated expected neural response during normative vs. hedonistic choice, using the DDM, for all trials in Dataset 1 (**d**), as well as separately for each condition in Dataset 2 (**e**) and Dataset 3 (**f**). Bottom (lIFG/aIns Results): Within the lIFG/aIns ROI associated with the DDM (blue voxels in **a–c**), BOLD response differed as predicted during normative choice vs. hedonistic choice when attributes conflict, in (**g**) Dataset 1 (N = 50) for all trials, as well as in (**h**) Dataset 2 (N = 49) and (**i**) Dataset 3 (N = 113) as a function of regulatory goals. Regulatory goals were designed to decrease hedonic weights and increase normative weights on choice (Dataset 2: Ethics, Partner; Dataset 3: Health) or to put weights on hedonic attributes (Dataset 3: Taste). Only conflict trials are considered across datasets. As predicted, normative choices activate the dACC, but only when goals result in a greater weight on hedonistic than normative attributes. Contexts that increase normative attribute weights yield less lIFG/aIns activity during normative choices (Dataset 2: Ethics, Partner; Dataset 3: Health). * Pp < 0.05, two-tailed; ** Pp < 0.01, two-tailed, based on one-sample t-tests against 0, or paired-tests between conditions. Error bars show standard error of the mean.

# Appendix 5

## Supplemental individual difference analyses for Datasets 2 and 3

The DDM predicts individual differences in the link between dlPFC activation and the relative weight given to normative vs. hedonic attributes. In particular, our model predicts that higher relative weight on virtuous vs. hedonic attributes should be *negatively* correlated with differences in dlPFC response when choosing virtuously vs. hedonistically under attribute conflict. In the main text, we presented evidence for just such a relationship in Dataset 1, which has the cleanest design for interpreting naturally occurring individual differences because it includes no instructed regulation component. Here, we report parallel analyses for the Natural conditions of Datasets 2 and 3.

## Effects of individual differences in the relative weighting of virtuous vs. hedonistic attributes in Dataset 2 (altruistic choice)

In Dataset 2, the proposals shown to participants were adjusted on an idiosyncratic basis to elicit generous choice in approximately 30% of trials. Moreover, decisions in an altruistic context are driven not only by weights on self and other, but also weight on inequality. Both of these facts might obscure individual differences in the relative weighting of normative (i.e., $w_{Other}$) vs. hedonic attributes (i.e., $w_{Self}$) in this dataset. We nevertheless analyzed the correlation between left dlPFC response on altruistic vs. selfish choices and the relative weights given to self vs. other considerations in the DDM in Natural Focus trials only. Although we did observe a negative correlation between these two variables, this relationship was not statistically significant ($r_{47}=-0.09$, $p=0.56$).

## Effects of individual differences in the relative weighting of virtuous vs. hedonistic attributes in Dataset 3 (food choice)

In Dataset 3, all participants were presented with similar choices in the Natural condition, permitting a cleaner test of the influence of naturally occurring individual differences in relative weighting of tastiness and healthiness on dlPFC response. We thus analyzed the correlation between left dlPFC response on healthy vs. unhealthy choices (conflict trials only) and the relative weights given to healthiness vs. tastiness considerations in the DDM in Natural Focus trials only. As predicted, we observed a significant negative correlation between these two variables ($r_{99}=-0.28$, $p=0.002$). Thus, participants who naturally used virtuous attributes more strongly than hedonic ones to guide their choices show *less* dlPFC activity. This result matches the finding in Dataset 1 and extends it to a different (non-social) choice domain.

