## [Editor Report]

This paper will be of interest to neuroscientists studying decision-making and the frontal lobe. On balance, the data provide more support for the view that the dorsolateral prefrontal cortex is involved in reading out the evidence in favor of different choice alternatives than the view that this region implements control processes that bias choices towards normative goals.

---

## [Decision Letter]

**Decision letter after peer review:**

Thank you for submitting your article "Evidence accumulation, not "self-control," explains dorsolateral prefrontal activation during normative choice" for consideration by *eLife*. Your article has been reviewed by 3 peer reviewers, one of whom is a member of our Board of Reviewing Editors, and the evaluation has been overseen by Michael Frank as the Senior Editor. The reviewers have opted to remain anonymous.

Essential revisions:

1) One concern is about potential circularity in the analyses of the dlPFC activity. Given that the ROI is identified via a model-based fMRI analysis with the anDDM predicted activity, isn't the current strategy of testing choice-related activity in this region (GLM2 analyses like Figure 4 d-f) a case of circular-analysis /double dipping (Kriegeskorte et al., 2009)? Wouldn't this bias the subsequent tests in this region of whether activity is higher for normative (generous, healthy) versus non-normative (selfish, tasty) choices towards the predictions of the anDDM? In this case, a stronger and more convincing test might be to show that these results hold in a region defined a priori based on previous studies arguing that dlPFC modulates attribute weights (e.g., Hare's work).

2) A second concern is how well the model is recoverable. It would be helpful for the authors to demonstrate parameter recovery, or point the reader to where those results can be found, if they were shown in prior work. (Regarding modeling quality control, the authors may want to consult the guidelines of Wilson and Collins, 2019).

(a) On a specific note, there is skepticism about the need for both input inhibition (v1-v2) as well as inhibition between the accumulators (zeta). How well is this zeta parameter recovered, particularly under minor to major misspecification of the drift rate function v? For instance, if v is in reality non-linear, or has an intercept (see above), might the currently mis-specified drift rate function result in what looks like a non-zero zeta?

(b) There is also a specific question of why the authors constrain zeta and γ in the model fits. By doing so they essentially guarantee concluding that there is self-excitation and mutual inhibition. Constraining parameter values shouldn't be necessary (though constraining the starting values of the search makes sense). All it does, potentially, is to turn a zero mean, high variance distribution into one with less variance but a biased mean.

3) A third question concerns the uniqueness of the neural predictions of the anDDM model. This should be clarified in the revision, specifically with regard to:

(a) Do the predictions of the anDDM differ from those based on a simpler notion of preference strength (e.g. Utility chosen option – utility unchosen option)? What about a conceptualization based in terms of default option/framing (Lopez-Persem et al., 2016)? Or would a model that focuses on conflict/choice difficulty/inverse confidence (Shenhav et al., 2014) make similar predictions? The authors may want to consider a more comprehensive model-comparison/model falsification approach (Palminteri et al., 2017) to validate their computational proposition at both the behavioral and neural activity levels.

(b) Do the predictions depend on specific modeling choices in the anDDM? For example, does a simpler attribute-weighted drift diffusion model (without the self-excitation and mutual inhibition components of the neural model) make similar predictions or not? Do the predictions depend on the specific linking function between model output and neural activity, since alternative formulations could be envisioned here (e.g. sum of the difference -rather than sum- of activity between the 2 neuron pools over time)?

(c) What about a simple account based on reaction time? What would happen, e.g. if RT is added as a parametric regressor (to the boxcar) and competed with the anDDM predictions?

(4) A fourth concern is about the strength of the neural evidence. The evidence on balance favors evidence accumulation, but perhaps not as strongly as one might hope. Where the evidence accumulation and attribute weighting/inhibition hypotheses make different predictions are in the conditions where regulatory focus is on normative goals – attribute weighting would predict greater activity for normative choices while evidence accumulation would predict greater activity for non-normative choices. Only one of the three tests here (health focus in food choice) provides straightforward support in one direction (favoring the evidence accumulation account). The other two tests (ethics and partner focus in altruistic choice) show no difference in activity, which isn't supportive of the attribute weighting account, but seems only weak evidence in favor of the evidence accumulation account. The test of whether regulatory instructions increase or decrease activity in dlPFC is similarly ambiguous, hardly supportive of the attribute weighting account, but only weakly supportive of the predictions of the evidence accumulation account. Furthermore, about half of the ROI critical tests (in Figure 4 e-f) are quite borderline (non-)significant (between P = 0.02 and 0.07). Overall, the strength of the neuroimaging evidence in favor of the proposed theory seems moderate at best.

(a) One response would be the authors to simply acknowledge these issues and temper their claims accordingly.

(b) Alternatively, multiple reviewers wondered if the authors could exploit individual differences to gather further support for their claims. For instance, for Study 3, in the Natural condition, could the authors only analyze subjects who overwhelmingly preferred tasty foods? Or, if the sample is large enough, the authors could look for brain-behavior correlations? Do subjects who show a larger behavioral weight asymmetry also show a larger DLPFC asymmetry?

5) Finally, the authors should address two specific critiques that might be raised regarding their manipulations in Studies 2 and 3. Several possibilities for how to address these were suggested, but beyond these specific suggestions, the essential thing is that authors somehow consider these critiques in the manuscript and rebut them.

(a) One potential critique of Study 2 is that subjects in the instructed conditions (focus on ethics, focus on partner) might take the instructions too literally and essentially be solving math problems rather than truly valuing the outcomes. This would make it less of a value-based decision-making task and more of a perceptual one. Presumably this would disengage the DLPFC, and the other regions. Could the authors comment on whether the standard "reward" regions, e.g. VMPFC, striatum, were active during these conditions? Is there similar evidence that the DLPFC, dorsal ACC, and insula reflect accumulated evidence in perceptual decisions? Are the absolute levels of activity (shown in Figure 4e) meaningful, in terms of saying whether the DLPFC was still engaged in these conditions? Another thing that would make the results from this study more compelling would be if the authors found a significant difference-in-differences between "Natural" and "Partner" conditions in DLPFC or other areas. This would help to alleviate concerns that brain activity is just noisier in the "Ethics" and "Partner" conditions.

(b) A potential critique of Study 3 is that the DLPFC is simply responding to disobeying the experimenters' instructions. Is there any way that the authors can rule this out? Could the authors use a more continuous analysis to show that decisions with a subjective value difference of zero show the most activity in DLPFC, while those with large subjective value differences, in either direction, show less activity? One particular reason for this concern is the lack of correspondence between the behavioral and fMRI data. The naDDM reveals nearly identical behavioral weights in the Natural and Taste conditions, and yet the DLPFC are very different. How can we make sense of this?

Friston, K.J., Penny, W.D., and Glaser, D.E. (2005). Conjunction revisited. NeuroImage 25, 661-667.

Kriegeskorte, N., Simmons, W.K., Bellgowan, P.S.F., and Baker, C.I. (2009). Circular analysis in systems neuroscience: the dangers of double dipping. Nat. Neurosci. 12, 535-540.

Lopez-Persem, A., Domenech, P., and Pessiglione, M. (2016). How prior preferences determine decision-making frames and biases in the human brain (*eLife* Sciences Publications Limited).

Palminteri, S., Wyart, V., and Koechlin, E. (2017). The Importance of Falsification in Computational Cognitive Modeling. Trends Cogn. Sci. 21, 425-433.

Shenhav, A., Straccia, M.A., Cohen, J.D., and Botvinick, M.M. (2014). Anterior cingulate engagement in a foraging context reflects choice difficulty, not foraging value. Nat. Neurosci. 17, 1249-1254.

Wilson, R.C., and Collins, A.G. (2019). Ten simple rules for the computational modeling of behavioral data. *ELife* 8, e49547.

[Editors' note: further revisions were suggested prior to acceptance, as described below.]

Thank you for resubmitting your work entitled "Evidence accumulation, not "self-control," explains dorsolateral prefrontal activation during normative choice" for further consideration by *eLife*. Your revised article has been evaluated by Michael Frank (Senior Editor) and a Reviewing Editor.

The reviewers appreciated your substantial and responsive revisions, but also identified some remaining issues that should be addressed, as outlined below:

*Reviewer #1 (Recommendations for the authors):*

The authors have comprehensively addressed the previous round of concerns and the revised manuscript is stronger and more compelling as a result.

*Reviewer #2 (Recommendations for the authors):*

The authors have done an admirable job addressing the review team's comments. They looked into parameter recovery and ultimately decided to go with a simpler model that yields similar conclusions. They also added a couple of datasets to Study 3 to help bolster those results.

My overall impression of the paper is fairly positive, though some of my previous reservations remain. To really convincingly make their point, the authors would have ideally shown a dataset where DLPFC activity is significantly higher for the hedonistic choice – an actual reversal of the phenomenon, rather than just elimination of it. This weakness is somewhat mitigated by the individual difference results, though I think that should be expanded on in the paper, since right now they only report it for Dataset 1, and they don't fully report the statistics for Datasets 2 & 3 in their replies.

I also thought that the very last section of the Results section was weak and could, maybe should, be removed. I think the authors have already demonstrated by this point in the paper that the activity in DLPFC is not simply responding to the condition. It's strange to report these main effects but then immediately dismiss them as likely being due to evidence accumulation. For that matter, the lack of effect in Dataset 2 could also be spuriously due to evidence accumulation canceling out an actual main effect. This last section is just messy and I don't see why it's helpful.

I thought that the reference list could be improved, given the focus of the paper. There are multiple mentions of value DDM and fMRI investigations of this, and yet there were no references to the value DDM work by Milosavljevic and Krajbich (2010) or the older work by Busemeyer and colleagues. There were also no references to the fMRI studies on value DDM by Gluth et al. (2012), Rodriguez et al. (2015), Pisauro et al. (2017), etc.

I also had some issues with a couple of the figures.

Figure 2 – I do not understand how the axes are coded. The dependent variable is %healthy choice, so you would think the axes would be the difference in tastiness and healthiness between the healthy and unhealthy options. However, both of those differences go from -3 to +3, indicating that that's not the case – the healthy option can't be less healthy than the unhealthy option. It seems to me that there should only be the top half of each of these figures. It's also unclear why these figures are mostly red, indicating a preference for healthy options, even when the weight on taste is higher. Is this due to the intercept in the model? If so, that would be useful to clarify. Finally, is panel d upside down? The y-axis says H-UH choice, and that should be most positive for the highest percentage of healthy choices, which should be for the light gray bar, not the black one, right?

Figure 4 – panels d and g need labels on the x-axis.

Appendix Figure 5 – to my eye this looks more like insula than IFG, but perhaps I'm mistaken.

Finally, I think it is important to include the Q-P plots in the supplements at least, just to make it clear that the model fits could be improved. I don't think this is critical to the paper, which doesn't really rely on the specific model, but the misfits are noticeable.

---

## [Author Response]

Essential revisions:1) One concern is about potential circularity in the analyses of the dlPFC activity. Given that the ROI is identified via a model-based fMRI analysis with the anDDM predicted activity, isn't the current strategy of testing choice-related activity in this region (GLM2 analyses like Figure 4 d-f) a case of circular-analysis /double dipping (Kriegeskorte et al., 2009)? Wouldn't this bias the subsequent tests in this region of whether activity is higher for normative (generous, healthy) versus non-normative (selfish, tasty) choices towards the predictions of the anDDM? In this case, a stronger and more convincing test might be to show that these results hold in a region defined a priori based on previous studies arguing that dlPFC modulates attribute weights (e.g., Hare's work).

We agree that it is important to demonstrate that the predicted pattern holds in independently defined ROIs that have been previously argued to be associated with self-control. Thus, we have replaced the original, circular analyses with a new set of analyses. In the main paper, we focus on an independent, functionally defined ROI in the left dlPFC based on a meta-analysis of the term ‘inhibitory control’ using the meta-analytic database NeuroQuery^1^ which has the advantage of using an ROI defined by hundreds of studies examining a particular term. Following suggestions of the reviewers and editorial team, we also report the same set of analyses in the supplemental materials in an ROI based on Hare’s seminal work. Specifically, we show results for a cluster in the left dlPFC region associated with healthy vs. unhealthy choices in dieters^2^. From each of these ROIs, we extracted the contrast of generous vs. selfish choice in Dataset 1 and 2, and the contrast of healthy vs. unhealthy choice during conflicted and unconflicted choices in Dataset 3.

The results confirmed the predictions of an evidence accumulation account, albeit with a small amount of heterogeneity across studies and ROIs. To give the reviewers a sense of this heterogeneity, we here give a high-level summary of results from each of these analyses. In Dataset 1 (Natural focus condition only), we found a significant increase in response in the dlPFC for generous vs. selfish choices in both ROIs (both *P* <.05). In Dataset 2 (Natural and Regulatory focus conditions), we found both a significant increase during generous vs. selfish choices in the Natural focus condition in both ROIs (both *P* <.04), as well as the predicted difference among conditions in ethics and other-focused conditions in both ROIs (both *P* <.004). In Dataset 3 (Natural and Regulatory focus conditions), we have now added in data from two additional dietary self-regulation studies for increased power (see below for further details). Here, we also found a significant increase in BOLD response during Natural trials in both ROIs (both P <.02). We also found a significant effect of condition on this difference in the NeuroQuery ROI (P = .02), as well as a marginal effect in the Hare ROI (P = .099). In both ROIs, this was driven by a significantly smaller difference between healthy and unhealthy choices in the Health Focus condition compared to the Natural Focus condition (P = .02 in the NeuroQuery ROI, P = .05 in the Hare ROI).

For full disclosure, we also note here that in our previous Study 3 (which is now part of Dataset 3), as we were performing these analyses, we found a small error in the coding of contrast images. Correction of this error changed some of the statistics and effects in our original Study 3 sample slightly, although it did not change the main conclusion. We have now double- and triple-checked all analyses to ensure that there are no further errors. To assure ourselves that our results were robust, we also sought to gain additional power by adding in data from two additional previously collected datasets on dietary self-regulation that had a nearly identical paradigm. Adding in these studies has strengthened some of the results, and reassured us that results are largely similar across samples.

2) A second concern is how well the model is recoverable. It would be helpful for the authors to demonstrate parameter recovery, or point the reader to where those results can be found, if they were shown in prior work. (Regarding modeling quality control, the authors may want to consult the guidelines of Wilson and Collins, 2019).(a) On a specific note, there is skepticism about the need for both input inhibition (v1-v2) as well as inhibition between the accumulators (zeta). How well is this zeta parameter recovered, particularly under minor to major misspecification of the drift rate function v? For instance, if v is in reality non-linear, or has an intercept (see above), might the currently mis-specified drift rate function result in what looks like a non-zero zeta?(b) There is also a specific question of why the authors constrain zeta and γ in the model fits. By doing so they essentially guarantee concluding that there is self-excitation and mutual inhibition. Constraining parameter values shouldn't be necessary (though constraining the starting values of the search makes sense). All it does, potentially, is to turn a zero mean, high variance distribution into one with less variance but a biased mean.3) A third question concerns the uniqueness of the neural predictions of the anDDM model. This should be clarified in the revision, specifically with regard to:(a) Do the predictions of the anDDM differ from those based on a simpler notion of preference strength (e.g. Utility chosen option – utility unchosen option)? What about a conceptualization based in terms of default option/framing (Lopez-Persem et al., 2016)? Or would a model that focuses on conflict/choice difficulty/inverse confidence (Shenhav et al., 2014) make similar predictions? The authors may want to consider a more comprehensive model-comparison/model falsification approach (Palminteri et al., 2017) to validate their computational proposition at both the behavioral and neural activity levels.(b) Do the predictions depend on specific modeling choices in the anDDM? For example, does a simpler attribute-weighted drift diffusion model (without the self-excitation and mutual inhibition components of the neural model) make similar predictions or not? Do the predictions depend on the specific linking function between model output and neural activity, since alternative formulations could be envisioned here (e.g. sum of the difference -rather than sum- of activity between the 2 neuron pools over time)?

We address this set of points together because in addressing Point 2 (regarding model quality control), we also addressed a number of the issues raised in Point 3 (i.e., do predictions differ in a simpler attribute-weighted DDM?).

First, because we had based our original anDDM on a model from a previously-published paper (Hare et al., 2011), we did not originally do the kind of model verification analyses that the reviewers very rightly pointed out are important to do. We have now conducted a number of analyses to assess the extent to which specific model parameters are identifiable/recoverable, as well as the extent to which our predictions depend on the specific choice of model (e.g. neural network DDM vs. a more canonical DDM, see point 3b). In doing so, we found that, while parameters related to more traditional measures common to both a DDM and a neural DDM (i.e., attribute weighting, threshold, non-decision time) showed excellent identifiability, the parameters related to lateral inhibition and auto-excitation did not. Below, we summarize these results, before discussing our approach to this issue in the revised manuscript.

Parameter identifiability

To address model recovery questions, we performed model recovery exercises on a superset anDDM model that includes several changes suggested by the reviewers, including adding a constant (intercept) term to the drift rate, allowing for three distinct attributes to influence value (as this is the maximum used for Datasets 1 and 2, while Dataset 3 has only two distinct attributes) and allowing recurrent inputs to take on both positive and negative values. Thus, we estimated 8 separate parameters.

Note that we did not allow mutual excitation (as suggested by the reviewers) as we found that doing so made the model estimates extremely unstable, and, as elaborated below, we think this does not make sense theoretically or conceptually. We then simulated 150 datasets with random values of parameters, and then computed the correlation between the estimated parameters and the true values. This in general showed high correlations with the fitted parameters of interest, which relate to weights on attributes. Recovery of each of three distinct attributes was excellent, with Pearson’s correlation coefficients ranging from r = .87 to r = .96. We also observed reasonably decent recovery of the drift constant, with a Pearson’s correlation coefficient of r = .62. Recovery of threshold value (r = .75) and non-decision times (r = .76) were also acceptable. However, we did not observe clean recovery of the recurrence and inhibition parameters (r = .14 and r = -.1129) respectively. This suggests that these parameters may not actually have been necessary for model fits. We thus turned to model comparison exercises to more formally assess this possibility.

Model comparison

We undertook a thorough set of analyses, based on suggested alternatives by reviewers, to determine which model provided the best fit to the data. Because of the time-consuming nature of model-fitting, and because of some issues identified in this process as well as the exercises described above on parameter recovery, we performed these exercises only in Dataset 3a (food choice with regulation) rather than in all datasets. We acknowledge that this is not a full test of the model in all datasets, and of course would be happy to finish these estimations and report them in greater detail in the supplements if the editor or reviewers think this would strengthen the work.

For selecting the best anDDM, we estimated models both with and without auto excitation, with and without lateral inhibition, and with either relative values as inputs to the two neuronal pools, or raw values as inputs. We also estimated a canonical DDM. All models fit a constant intercept in the drift, specifying a default preference towards or away from accepting the proposal.

On point 2b regarding constraints on parameters: while we agree with the reviewers that a constraint on the auto-excitation parameter should not be necessary, we believe there are good conceptual reasons not to allow the lateral inhibition parameter to take on values that entail excitation rather than inhibition. In particular, the distinction between a pool that favors one response and a pool that favors a competing response is lost if the two pools mutually excite each other, and we know of no extant models in the literature that have ever suggested mutual excitation of neuronal populations for competing responses. Thus, in the models reported below, we opted to retain constraints on this parameter when performing model-fitting exercises.

Author response table 1 summarizes the average AIC/BIC value across participants for each model. The best model according to the AIC/BIC for each condition is highlighted in bold.

**Author response table 1. sa2table1:** Model fits as assessed by AIC and BIC.

Model	Recurrence	Lateral Inhibition	RelativeValues	NaturalAIC/BIC	TasteAIC/BIC	HealthAIC/BIC
1	x	x	x	1401/1418	1397/1415	1418/1435
2		x	x	1403/1418	1395/1410	1414/1430
3	x		x	1396/1411	1394/1409	**1392/1407**
4			x	**1395/1407**	**1388/1400**	1409/1421
5_	x	x		1436/1453	1409/1427	1436/1453
6	x			1428/1443	1420/1435	1445/1460
7		x		1417/1432	1403/1418	1436/1451
8				1429/1442	1419/1432	1445/1457
9	Canonical DDM	1397/1409	1393/1406	1412/1425		

As can be seen, the neural DDM with no recurrence and no lateral inhibition (Model 4 in Author response table 1) provides the best fit to the data in two conditions, and the model with recurrence but no lateral inhibition provides the best fit in a third. Models with raw inputs into the accumulators consistently performed worse than their relative value counterparts. Thus, for purposes of the paper, we now report results of Model 4 of the anDDM (no lateral inhibition, no recurrence) in the supplemental material.

However, examining model convergence using the Gelman-Rubin r-hat statistic suggested that although Model 4 generally had the lowest AIC/BIC, it also occasionally failed to converge on a stable set of parameters for a fraction of subjects (on average ~6% across the conditions), as evidenced by R-hat values > 1.1, the conventional threshold for assessing good model convergence. The standard DDM showed no such issues, with all parameters for all subjects showing excellent convergence (all R-hat values < 1.003). Moreover, in general, the canonical DDM had AIC/BIC values similar to, though generally slightly higher than, the best-fitting neural DDM. The DDM was the 2^nd^-best fit according to both AIC and BIC in two conditions, and third-best fit in the third.

Given these issues with model convergence and parameter recovery, and the fact that the canonical DDM ran a close second in terms of model BIC values, we turned to this version and asked whether it would account for the same findings as the anDDM, as suggested by the reviewers and editor (point 3b). In brief, we found that a simple DDM reproduces all of the key predictions made by the anDDM regarding activity during normative vs. hedonistic choice, and identifies similar neural areas in all three studies (see our main paper for these results). Importantly, we also found that the attribute weighting parameters of the DDM correlated quite highly with the attribute weighting parameters of the best-fitting anDDM (Model 4, all r >.97), giving us confidence that inferences about how these weights change with attention, and how these weights correlate with neural and behavioral responses, are robust to alternative model specifications.

We thus have opted to revise our paper to focus on the DDM as a canonical evidence accumulation model. This has the advantage of making the paper more accessible to a readership who might be more familiar with the DDM, and requiring less description to introduce details of the model. We believe this move significantly streamlines the paper, while leaving our core message intact. However, we also included the winning anDDM results as a supplemental analysis to show that results are not specific to any precise instantiation of an evidence accumulation account. To further emphasize this, we revised wording of the introduction and discussion of the paper to make clearer that we think it is a general principle of evidence accumulation models, rather than any specific model, which produces the relationship we observe here between asymmetry in weighting of hedonic and normative attributes and differences in response on hedonistic vs. normative choices. We thus thank the reviewers for making these suggestions and encouraging us to take a more careful look at our modeling approach, as it has resulted in a much cleaner and clearer exposition of our logic and theory.

Alternative model specifications for the neural data

Regarding the questions raised in part 3a (alternative specifications to an evidence accumulation model, defined either as the DDM or anDDM), we have undertaken several additional analyses to address the reviewer’s suggestions. We performed these analyses only in Dataset 2 (altruistic choice with regulation), since our initial analysis even in this single dataset suggested that none of these alternative suggestions capture activation in the left dlPFC at even extremely low thresholds (P <.01, uncorrected). We elaborate on this conclusion below.

First, as suggested, we asked whether the utility of the chosen option – utility of the unchosen option provided a similar and/or better account of neural activity in the dlPFC or other regions. To do this we computed the utility of each option based on its integrated weighted attribute value (using estimated weights from the DDM). We then computed the chosen – unchosen value as a parametric regressor and estimated neural response correlated with this regressor. However, we failed to find any evidence that this regressor explained activation in the left dlPFC, even at a fairly liberal P <.01, uncorrected. We did observe some hints that Unchosen – Chosen activated the dorsomedial prefrontal cortex at liberal thresholds (P <.01, uncorrected), as has sometimes been reported in the literature, but this regressor was not as powerful a predictor as the DDM, which predicted activation in the dmPFC at whole-brain cluster corrected levels.

Second, we asked whether activity in these regions correlated with confidence (coded as the absolute value of the behavioral preference coded from strong yes to strong no, i.e. both strong yes and strong no indicated high confidence, weak yes and weak no indicated low confidence). Although we did observe a correlation with confidence in the occipital cortex and areas of motor cortex, this analysis also failed to identify any activation in the dlPFC or other key areas of interest even at P <.01, uncorrected.

Unfortunately, our experimental designs were not set up to directly address the question of whether a default option/framing explanation might account for the data. In our studies, the same default option was presented across all trials in all studies. Thus, we cannot fit neural activity that varies as a function of the value of the default option, as was done in Persem-Lopez (2016). However, we think it is unlikely that our results in the dlPFC are explained by a default bias. First, it is unclear why a default bias would be stronger on trials predicted by the DDM to require more vs. less evidence accumulation, so it is unclear how such an account would predict variation in the dlPFC of the sort that we see. Second, in Persem-Lopez et al. (2016), the areas sensitive to the default value lay outside the dlPFC area that we here associate with predictions of the DDM. Thus, we think that an evidence accumulation account is currently the best explanation of the results we see in the dlPFC. To reduce the working memory burden on the reader, we have opted not to include these additional analyses in the current version of the paper, which already has a larger number of results. However, we would be happy to add these results if the editor or reviewers feel it would substantially improve the paper. We also note an advantage of *eLife* over other journals, which is that the response to reviews is accessible to the public, so readers can if they desire find this information here in our response letter.

(c) What about a simple account based on reaction time? What would happen, e.g. if RT is added as a parametric regressor (to the boxcar) and competed with the anDDM predictions?

A model in which RT is included as a parametric modulator and allowed to compete with the DDM does indeed provide an excellent account of neural activity in several areas, including the left dlPFC area that we focus on in this paper, in part because the correlation between RT and predicted DDM activity is generally >.95. Indeed, we observe a three-way conjunction of activation correlated with RT in the left dlPFC, which overlaps to a large extent with the area identified in Hare et al. (2009)^2^ and our NeuroQuery ROI, when each individual study map is thresholded at P <.005, uncorrected. Activity associated with the DDM in the dlPFC generally does not continue to be significant after controlling for RT. However, we point out that the DDM regressor that we use will necessarily be a rough estimate of expected neural activity, because it is the average over multiple simulated trials with multiple RTs ranging around but not necessarily identical to the RT on the trial. RT, in contrast, is a single number that captures the time it took for the evidence accumulation process to actually cross the threshold on *that* trial. Thus, RT might in some cases be more strongly correlated with the neural activity than a rough estimate of that activity based on the DDM. It thus should not be wholly surprising that it provides as good or better accounts of activity in these regions.

Moreover, our basic point (that the dlPFC area often associated with self-control might be more reflective of evidence accumulation-related mechanisms than of value modulation) stands regardless of whether the precise computational function of this region is strictly related to evidence accumulation or instead related to RT. Indeed, our simulations show that RT should show the same pattern as neural response (see Figure 2—figure supplement 1), and in fact might even better reflect the actual evidence accumulation process than averaged DDM response over multiple trials (see Supplemental Analyses, bottom of pg. 2 to top of pg. 3). Regardless, then, of the precise computation served by the dlPFC, we should observe the changes in normative vs. hedonistic response patterns that are predicted by our model. We include a brief discussion of this issue in the main paper on pg. 11, Lines 206-210: “Note, however, that finding a correlation within this region could occur because the dlPFC performs the precise computations carried out by the DDM, or could occur if the dlPFC performs separate computational functions that activate proportionally to evidence accumulation activity (e.g., response time). In either case, we would expect the trial-by-trial activity of the dlPFC to correlate with predictions of the DDM.”

4) A fourth concern is about the strength of the neural evidence. The evidence on balance favors evidence accumulation, but perhaps not as strongly as one might hope. Where the evidence accumulation and attribute weighting/inhibition hypotheses make different predictions are in the conditions where regulatory focus is on normative goals – attribute weighting would predict greater activity for normative choices while evidence accumulation would predict greater activity for non-normative choices. Only one of the three tests here (health focus in food choice) provides straightforward support in one direction (favoring the evidence accumulation account). The other two tests (ethics and partner focus in altruistic choice) show no difference in activity, which isn't supportive of the attribute weighting account, but seems only weak evidence in favor of the evidence accumulation account. The test of whether regulatory instructions increase or decrease activity in dlPFC is similarly ambiguous, hardly supportive of the attribute weighting account, but only weakly supportive of the predictions of the evidence accumulation account. Furthermore, about half of the ROI critical tests (in Figure 4 e-f) are quite borderline (non-)significant (between P = 0.02 and 0.07). Overall, the strength of the neuroimaging evidence in favor of the proposed theory seems moderate at best.(a) One response would be the authors to simply acknowledge these issues and temper their claims accordingly.

We have undertaken several approaches to address this point. First, to increase statistical power in Dataset 3 (where some of our previously-report P-values were more modest), we have added in results from two studies with nearly identical design to our own, first testing that results are not significantly different across studies, and then combining them to analyze aggregate responses. This has resulted in P-values for the critical tests in our ROI ranging from P = .004 to P = .02 in the revised version of the manuscript. Second, we now also focus on the difference-of-differences comparison (i.e., the *change* in the contrast of normative vs. hedonistic choices across conditions, as suggested in a comment below), which we view as the more critical test of our model (consistent with reviewer suggestions in point 5a). In particular, we show it is not always clear based on the actual weights on normative and hedonic parameters that participants achieved whether one would expect a full reversal of effects in normatively-focused conditions (i.e., a significantly greater response for hedonistic compared to normative choice). Instead, our much stronger prediction (and one that is born out in *all* datasets at conventional P-values in our region-of-interest) is that a normative focus should significantly *decrease* this difference, compared to typical or hedonically focused choice contexts. This is the critical test that distinguishes our model from models of self-control that argue that going against default preferences should make normative choices *harder* and thus should activate the dlPFC *more* (assuming it represents an effortful self-control mechanism). The test of difference-of-differences is also significant at conventional P-values in all three datasets. Thus, while we agree with the reviewer and editors that our statistical evidence varies in its strength from study to study, we believe that the consistency of effects across all three datasets nevertheless provides relatively strong evidence *against* previous models, and moderate to robust evidence *for* an evidence accumulation account.

Moreover, we further emphasize that none of our three paradigms was directly designed to address precisely the question we examine here, and thus each has some issues that, on their own, likely reduce the statistical significance of our effects and add noise to observed response. We hope, however, that replication of our effects over three different paradigms and two different choice contexts with a combined total of over 200 subjects helps to allay worries that any individual test of our hypothesis either does or does not meet an arbitrarily defined threshold for significance of P <.05.

We also now seek to quantify more rigorously what the predictions of the DDM actually are, by running simulated activity for normative vs. hedonistic choices in each study for each individual, given the fitted parameters. These generally show that the predictions of our model correspond to observed activity. There are, however, a few cases where this is not true. One is the prediction that the Focus on Partner condition should evoke a full reversal of activation differences on generous vs. selfish choices in the dlPFC, and should be significantly different from the Ethics condition. We acknowledge that our results do not find this predicted effect, although it is somewhat difficult to know whether this might be due to noise, since the Focus on Partner condition evoked extremely high levels of generosity in some subjects, and thus reduced the number of selfish choice trials.

Nevertheless, we think it important to acknowledge these limitations, and appreciate the opportunity/suggestion from the reviewers and editors to take this approach. We now more directly address these issues on pg. 28-29 of the revised manuscript, writing, “The correspondence between predictions of the DDM and activation patterns in the dlPFC makes it tempting to conclude that this region performs this computational function. While this hypothesis is consistent with results from single-cell recordings, we also acknowledge that the dlPFC has been associated with many computational functions and roles, not all of which are mutually incompatible. Thus, it is possible that the dlPFC region observed here performs some sort of process that is correlated with, but not identical to, the neuronal computations of the DDM. Indeed, although we found that dlPFC responses mostly conformed to predictions of the DDM, we also found a few notable divergences. For example, during unconflicted trials, the DDM predicts that *hedonistic* choices should be strongly associated with greater activity than normative choices. While we observed a weak effect in this direction in Dataset 3 (the only dataset in which this prediction could be tested), it was not significant. Similarly, in Dataset 2, the DDM predicted that the comparison of generous vs. selfish choices should be lower when focused on one’s partner compared to when focusing on ethics, while the empirical data showed no difference. While some of these discrepancies could be due simply to the noisiness inherent to fMRI (especially in studies that were originally designed to address other questions), it is also possible that they serve as clues that the computational function of the dlPFC may be strongly correlated with, but importantly distinct from, the evidence accumulation process itself. Future work, including computational modifications or additions to the DDM, and recordings from other modalities like EEG or intracranial recording, may help to elucidate the precise computational functions served by this area and the ways in which it promotes adaptive choice and normative behavior.”

(b) Alternatively, multiple reviewers wondered if the authors could exploit individual differences to gather further support for their claims. For instance, for Study 3, in the Natural condition, could the authors only analyze subjects who overwhelmingly preferred tasty foods? Or, if the sample is large enough, the authors could look for brain-behavior correlations? Do subjects who show a larger behavioral weight asymmetry also show a larger DLPFC asymmetry?

This is an interesting suggestion, as the DDM does predict that individual differences in the asymmetry between hedonic and normative attributes should correlate with the difference between hedonistic and normative choice. Indeed, we find some evidence for the predicted effects across individuals. In Dataset 1 (previously Study 1), the difference in dlPFC response on generous vs. selfish choices is correlated *negatively* with the asymmetry between weight on other vs. weight on self in the NeuroQuery ROI (*r =* -.31, *P* = .04). We observed similar results in Dataset 2 and 3. In Dataset 2 (previously Study 2), a mixed effects regression with BOLD response on generous vs. selfish trials in the NeuroQuery ROI as the dependent variable, the difference in weight on other vs. weight on self in each condition as a fixed effect, and participant as a random intercept revealed a significant negative association (*P* = .02), although this dropped to non-significance when the effect of condition as a main effect was also included. In Dataset 3, a mixed effects regression with BOLD response on healthy vs. unhealthy choices as the dependent variable, the difference in weights on health vs. weights on taste as a fixed effect, and participant as a random intercept likewise revealed a significant negative association (*P* = .01). Unlike in Dataset 2, there was no significant effect of condition after accounting for weights on taste and health. Thus, we think evidence from individual difference analyses across all three studies largely supports our theory. We do note that Dataset 2 showed the weakest evidence for our cases, since the effect of individual differences across participants and conditions was non-significant after accounting for condition. However, we also note that Dataset 2 was the only study in which we gave each participant different offers, designed to achieve roughly similar amounts of generosity across all participants. This would have the effect of weakening the effect of individual differences in this study. Due to these complexities, we have currently opted only to report individual difference results for Dataset 1 in the revised manuscript, which has the cleanest design and interpretation for studying the relationship between inherent preference weights and observed BOLD response (see pg. 15, lines 292-294). We can include similar analyses for Dataset 2 and 3, either in the main manuscript or in supplemental analyses, if the editor or reviewers believe it is important to do so, and thank the reviewers for this suggestion.

5) Finally, the authors should address two specific critiques that might be raised regarding their manipulations in Studies 2 and 3. Several possibilities for how to address these were suggested, but beyond these specific suggestions, the essential thing is that authors somehow consider these critiques in the manuscript and rebut them.(a) One potential critique of Study 2 is that subjects in the instructed conditions (focus on ethics, focus on partner) might take the instructions too literally and essentially be solving math problems rather than truly valuing the outcomes. This would make it less of a value-based decision-making task and more of a perceptual one. Presumably this would disengage the DLPFC, and the other regions. Could the authors comment on whether the standard "reward" regions, e.g. VMPFC, striatum, were active during these conditions?

We concur that it is important to know whether participants are simply solving math problems in Datasets 2 and 3, or instead are actually calculating values. We have several lines of evidence to support the idea that they are actually computing values rather than simply performing abstract cognitive operations. First, we have previously reported that we observe multivariate encoding of attribute values for *all* attributes in both tasks (i.e., $Self, $Other, and inequality in Dataset 2, Healthiness and Tastiness in Study 3a of Dataset 3) and the integrated decision value in an area of the vmPFC typically thought to encode values^3^. Second, we also observe a univariate correlation with overall decision value (i.e. Strong No to Strong Yes) in the vmPFC in all three conditions for Dataset 2 (P <.05, uncorrected within each condition) and in Dataset 3 (P <.05, uncorrected within each condition). Taken together, we think this indicates that participants are indeed computing values, as opposed to simply solving math problems.

We now include a footnote in the Discussion section that references our prior results in the discussion and addresses this critique on pg. 27: “An objection could be raised that the results in Datasets 2 and 3 do not arise from changes in evidence accumulation of value-related signals, but are due instead to demand effects that result in disengagement of typical valuation circuitry during choice, and thus disengagement of the dlPFC. Yet previous analyses on these datasets^1,19^ suggest that the vmPFC and other value-related regions, including a value-coding region of the dlPFC, continue to encode both individual attribute values and integrated values in both regulated and unregulated conditions. This suggests that our results are not simply due to disengagement of value computations.”

Is there similar evidence that the DLPFC, dorsal ACC, and insula reflect accumulated evidence in perceptual decisions?

There is some evidence that the dlPFC in particular might be important for accumulating evidence in perceptual decisions. For example, regions of the dlPFC that are nearly identical to the region we observe here correlating with our value-based DDM predictions (as well as areas of ACC and insula/inferior frontal gyrus) are also more active on hard compared to easy trials in perceptual decision-making tasks, consistent with an evidence accumulation account^4-6^, although we were not able to find any studies that specifically model the trial-by-trial predicted activity of a DDM to identify the dlPFC in a perceptual task. We have included these additional references in the introduction of the paper.

Are the absolute levels of activity (shown in Figure 4e) meaningful, in terms of saying whether the DLPFC was still engaged in these conditions? Another thing that would make the results from this study more compelling would be if the authors found a significant difference-in-differences between "Natural" and "Partner" conditions in DLPFC or other areas. This would help to alleviate concerns that brain activity is just noisier in the "Ethics" and "Partner" conditions.

We agree with the reviewer that the difference in activity between normative vs. hedonistic choice may be more informative than the absolute levels. We thus have revised Figure 4 to show the differences between normative and hedonistic choices, and to highlight the “difference of differences” effects that are most important for our theory. We have also highlighted these differences in the main text. As can be seen from both the main text and the figure, it is these difference-in-differences which is significantly different across conditions, more so than the differences in absolute overall values. Thus, we confirm that there is a significant difference-in-differences between Natural and Partner (and Natural and Ethics) in the altruistic choice task, and a significant difference-in-differences between Natural and Health Focus conditions in the dietary decision-making task.

(b) A potential critique of Study 3 is that the DLPFC is simply responding to disobeying the experimenters' instructions. Is there any way that the authors can rule this out? Could the authors use a more continuous analysis to show that decisions with a subjective value difference of zero show the most activity in DLPFC, while those with large subjective value differences, in either direction, show less activity? One particular reason for this concern is the lack of correspondence between the behavioral and fMRI data. The naDDM reveals nearly identical behavioral weights in the Natural and Taste conditions, and yet the DLPFC are very different. How can we make sense of this?

We agree that there are some puzzling divergences between the observed dlPFC response and the expected predictions of the DDM (and anDDM), although the stark difference between the natural and taste conditions has now disappeared after adding in additional data and correcting a small error in the coding of contrast images. We also think it unlikely that the dlPFC is simply responding to disobeying the experimenter’s instructions, since this would not predict different patterns of results for conflicted vs. unconflicted trials (a pattern we observe in the data).

We also note that the analysis requested by the reviewer (showing that decisions with a subjective value difference of zero show the most activity in the dlPFC) is essentially what we already show in our parametric analysis, showing the dlPFC correlates parametrically with predictions of the DDM, which has higher predicted activity for less extreme values regardless of sign, and lower predicted activity for more extreme values.

However, to more thoroughly address this concern, we also sought other ways to understand the predictions of the model, which we now report in the paper. In particular, we now explicitly use the model to make predictions about how activation in different regulatory conditions should compare, taking into account differences not only in attribute weights, but also overall drift and threshold differences. We show that in some cases counterintuitive predictions results (i.e., the difference between healthy and unhealthy choices in the taste condition is actually predicted by simulation to be somewhat *lower* than in the natural condition, which is also what is observed in our data after combining several datasets).

References

1. Dockès, J.m., et al. (2020). eLife, 9, e53385.

2. Hare, T.A., Camerer, C.F. & Rangel, A. (2009). Science, 324, 646-648.

3. Tusche, A. & Hutcherson, C.A. (2018). eLife, 7, e31185.

4. Heekeren, H., Marrett, S., Bandettini, P. & Ungerleider, L. (2004). Nature, 431, 859-862.

5. Liu, T. & Pleskac, T.J. (2011). Journal of neurophysiology, 106, 2383-2398.

6. Pedersen, M.L., Endestad, T. & Biele, G. (2015). Plos One, 10, e0140361.

7. Plassmann, H., O'Doherty, J. & Rangel, A. (2007). Journal of Neuroscience, 27, 9984-9988.

8. Teoh, Y.Y., Yao, Z., Cunningham, W.A. & Hutcherson, C.A. (2020). Nature Communications, 11, 1-13.

9. Baumgartner, T., Knoch, D., Hotz, P., Eisenegger, C. & Fehr, E. (2011). Nature Neuroscience, 14, 1468-1474.

10. Ruff, C.C., Ugazio, G. & Fehr, E. (2013). Science, 342, 482-484.

11. Hare, T.A., Malmaud, J. & Rangel, A. (2011). The Journal of Neuroscience, 31, 11077-11087.

12. Maier, S.U., Makwana, A.B. & Hare, T.A. (2015). Neuron, 87, 621-631.

[Editors' note: further revisions were suggested prior to acceptance, as described below.]

Reviewer #2 (Recommendations for the authors):The authors have done an admirable job addressing the review team's comments. They looked into parameter recovery and ultimately decided to go with a simpler model that yields similar conclusions. They also added a couple of datasets to Study 3 to help bolster those results.My overall impression of the paper is fairly positive, though some of my previous reservations remain. To really convincingly make their point, the authors would have ideally shown a dataset where DLPFC activity is significantly higher for the hedonistic choice – an actual reversal of the phenomenon, rather than just elimination of it. This weakness is somewhat mitigated by the individual difference results, though I think that should be expanded on in the paper, since right now they only report it for Dataset 1, and they don't fully report the statistics for Datasets 2 & 3 in their replies.

We thank the reviewer for their positive assessment of the changes that we have made, and their continued suggestions for improvement. We fully agree that it would have been more compelling to show a complete and significant reversal of effects, as we point out in the discussion in lines 603-618. We hope that our discussion of why we might observe divergences from some model predictions will spur further research on this topic.

We also agree with the reviewer that including a little more detail on the individual difference analyses for Studies 2 and 3 is potentially useful, although, as we pointed out in our response to the previous round of reviews (and elaborate further on in the new supplementary Results section), we do think there are some important caveats to being able to fully interpret these results. Thus, we have opted to present these results in a supplementary analysis in Appendix 5, focusing primarily on individual differences in Natural trials (as suggested by Reviewer 1).

I also thought that the very last section of the Results section was weak and could, maybe should, be removed. I think the authors have already demonstrated by this point in the paper that the activity in DLPFC is not simply responding to the condition. It's strange to report these main effects but then immediately dismiss them as likely being due to evidence accumulation. For that matter, the lack of effect in Dataset 2 could also be spuriously due to evidence accumulation canceling out an actual main effect. This last section is just messy and I don't see why it's helpful.

We fully agree and thank the reviewer for drawing our attention to this issue. We have now removed this section from the paper and references to it in the discussion. We believe that these revisions have increased clarity and streamlined the results.

I thought that the reference list could be improved, given the focus of the paper. There are multiple mentions of value DDM and fMRI investigations of this, and yet there were no references to the value DDM work by Milosavljevic and Krajbich (2010) or the older work by Busemeyer and colleagues. There were also no references to the fMRI studies on value DDM by Gluth et al. (2012), Rodriguez et al. (2015), Pisauro et al. (2017), etc.

We agree that we should have included these references in the paper, since they are important and help to inform both the model and the kinds of regions we might expect to correlate with evidence accumulation. We now include references to the papers/work mentioned by the reviewer where appropriate.

I also had some issues with a couple of the figures.Figure 2 – I do not understand how the axes are coded. The dependent variable is %healthy choice, so you would think the axes would be the difference in tastiness and healthiness between the healthy and unhealthy options. However, both of those differences go from -3 to +3, indicating that that's not the case – the healthy option can't be less healthy than the unhealthy option. It seems to me that there should only be the top half of each of these figures. It's also unclear why these figures are mostly red, indicating a preference for healthy options, even when the weight on taste is higher. Is this due to the intercept in the model? If so, that would be useful to clarify. Finally, is panel d upside down? The y-axis says H-UH choice, and that should be most positive for the highest percentage of healthy choices, which should be for the light gray bar, not the black one, right?

The axes represent the difference in healthiness/tastiness between the proposed option and the default option, not between the healthy and unhealthy option. So, it is possible for the proposed option to be both more or less healthy, and more or less tasty, than the default. This is also why the colors are largely red for two of the quadrants: when the proposed option is both healthier *and* tastier than the default (upper right-hand quadrant), the model almost always chooses it (i.e., makes the healthier choice). Similarly, when the proposed option is both less healthy *and* less tasty (lower left-hand quadrant), the model almost always rejects it (also the healthier choice). We have clarified this by adding the following text to the figure caption:

“Heat maps display simulated healthy/unhealthy choices for binary choices consisting of a proposed option vs. a default option. Both options are characterized by attribute values such as the perceived tastiness and healthiness of the two food options, with plot axes capturing the relative value difference in tastiness (x-axis) and healthiness (y-axis) of the proposal vs. the default. For example, for each heat map, the upper right-hand quadrant shows likelihood of a healthy choice when the proposed option was both tastier and healthier than the default. The upper left-hand quadrant illustrates the likelihood of a healthy choice when proposed foods were healthier but less tasty than the default and so forth. Warmer colors indicate a higher likelihood of choosing the healthier option.”

We also note that panel d’s axis is not upside down. This figure displays the predicted difference in activity of the accumulator for healthy vs. unhealthy choices when tastiness and healthiness do *not* conflict (left bars) vs. when they do (right bars). In cases of no attribute conflict (upper right-hand quadrant and lower left-hand quadrant in Figure 2 a), the model predicts that healthy choices (compared to unhealthy choices) should result in *lower* accumulator response. We have thus opted not to alter the figure or figure caption, since we have described this important result in both the main text and the figure caption. However, if the reviewer or editor feels additional text would help clarify this, we would be very happy to take suggestions!